# Neutrophil-derived catecholamines mediate negative stress effects on bone

Miriam E. A. Tschaffon-Müller[1,10], Elena Kempter [2,10], Lena Steppe [1], Sandra Kupfer[2], Melanie R. Kuhn[1], Florian Gebhard[3], Carlos Pankratz[3], Miriam Kalbitz[3,4], Konrad Schütze[3], Harald Gündel[5], Nele Kaleck[5], Gudrun Strauß[6], Jean Vacher[7,8], Hiroshi Ichinose[9], Katja Weimer [5], Anita Ignatius[1], Melanie Haffner-Luntzer [1,11] & Stefan O. Reber [2,11] ✉

Mental traumatization is associated with long-bone growth retardation, osteoporosis and increased fracture risk. We revealed earlier that mental trauma disturbs cartilage-to-bone transition during bone growth and repair in mice. Trauma increased tyrosine hydroxylase-expressing neutrophils in bone marrow and fracture callus. Here we show that tyrosine hydroxylase expression in the fracture hematoma of patients correlates positively with acknowledged stress, depression, and pain scores as well as individual ratings of healing-impairment and pain-perception post-fracture. Moreover, mice lacking tyrosine hydroxylase in myeloid cells are protected from chronic psychosocial stress-induced disturbance of bone growth and healing. Chondrocyte-specific β2-adrenoceptor-deficient mice are also protected from stress-induced bone growth retardation. In summary, our preclinical data identify locally secreted catecholamines in concert with β2-adrenoceptor signalling in chondrocytes as mediators of negative stress effects on bone growth and repair. Given our clinical data, these mechanistic insights seem to be of strong translational relevance.

Chronic psychosocial stress is a risk factor for a plethora of psychosomatic disorders, including depression and posttraumatic stress disorder (PTSD)[1]. These diseases have been linked to long-bone growth retardation, osteoporosis and increased bone fracture risk in a number of clinical studies[2–7]. A reduced bone mass has been also revealed in murine stress models resulting in a depressive-like phenotype, e.g. the chronic mild stress paradigm[8–12]. In contrast to mouse models for depression, employing the chronic subordinate colony housing (CSC)

paradigm as an acknowledged model for social stress-associated PTSD in male mice[13,14], we showed earlier that mental traumatization in adolescent mice negatively impacts cartilage-to-bone transition during endochondral ossification in the epiphyseal growth plate, the main site of longitudinal growth of the long bones, while appositional bone growth seems to be undisturbed[15]. In detail, CSC mice show reduced tibia and femur lengths, mineral deposition at the growth plate and Runt-related transcription factor 2 (Runx2) expression in hypertrophic

[1]Institute of Orthopaedic Research and Biomechanics, Ulm University Medical Center, Ulm, Germany. [2]Laboratory for Molecular Psychosomatics, Department of Psychosomatic Medicine and Psychotherapy, Ulm University Medical Center, Ulm, Germany. [3]Department of Orthopedic Trauma, Hand-, Plastic- and Reconstructive Surgery, Ulm University Medical Center, Ulm, Germany. [4]Department of Trauma and Orthopedic Surgery, University Hospital Erlangen, Friedrich-Alexander University Erlangen-Nuremberg, Erlangen, Germany. [5]Department of Psychosomatic Medicine and Psychotherapy, Ulm University Medical Center, Ulm, Germany. [6]Department of Pediatrics and Adolescent Medicine, Ulm University Medical Center, Ulm, Germany. [7]Department of Medicine, Institut de Recherches Cliniques de Montréal, Montréal, QC, Canada. [8]Institut de Recherche Cliniques de Montréal, Department of Medicine, Université de Montréal, H2W 1R7 Montréal, QC, Canada. [9]School of Life Science and Technology, Tokyo Institute of Technology, Yokohama, Japan. [10]These authors contributed equally: Miriam E. A. Tschaffon-Müller, Elena Kempter. [11]These authors jointly supervised this work: Melanie Haffner-Luntzer, Stefan O. Reber. ✉e-mail: stefan.reber@uni-ulm.de

chondrocytes in the growth plate, while growth plate and trabecular thickness as well as bone mineral density (BMD) are increased in CSC compared to single-housed control (SHC) mice[15]. An enhanced tyrosine hydroxylase (TH) expression, which is the rate limiting enzyme in catecholamine (CA) synthesis[16], in bone marrow (BM) cells located at the growth plates of CSC mice suggests that local CA signaling is involved in the negative CSC effects on bone metabolism[15]. Of note in this context, norepinephrine (NE) release by fibers of the sympathetic nervous system during chronic variable stress activates β3-adrenoceptor signalling in bone marrow niche cells and, consequently, reduces their CXCL12 secretion. This in turn increases hematopoietic stem cell proliferation and the release of neutrophils and inflammatory monocytes[17]. In a follow-up study we extended these findings by revealing that CSC mice undergoing standardized femur fracture show a delayed bone healing, again accompanied by a compromised cartilage-to-bone transition. Importantly, while endochondral ossification during fracture healing mimics the endochondral ossification process occurring at the growth plates during long bone growth, a higher percentage of osteoblasts is derived from chondrocytes in the fracture callus, aggravating the negative effects of disturbed chondrocyte-to-osteoblast transdifferentiation. Furthermore, CSC mice were characterized by a misbalanced inflammatory response in the fracture hematoma[18]. The latter was indicated by increased numbers of TH expressing neutrophils, and both delayed fracture healing and hematoma invasion of TH expressing neutrophils were prevented in CSC mice by injection with an unspecific β-adrenoceptor blocker prior to fracture surgery[18]. Together, our preclinical findings support the overall hypothesis that CSC-induced TH expression in myeloid cells and the subsequent local release of CAs compromises cartilage-to-bone transition during growth plate endochondral ossification and fracture healing.

## Results

### TH in the hematoma is linked to mental health in fracture patients

To assess physical and mental health as well as trauma load in male and female patients suffering upper ankle fracture, inpatients were asked to complete a number of established and standardized questionnaires on possible psychosomatic disorders such as somatic symptoms disorder, depression, anxiety, and psychosocial stress load (scales of the Patient Health Questionnaire, PHQ), social functioning and pain disability (Short Form Health Survey, SF36), and childhood adversity (Childhood Trauma Questionnaire, CTQ) on the days after the surgery as well as 3, 6, 9 and 12 months post surgery. During orthopedic surgery, the fracture hematoma was removed and analyzed for TH protein expression as well as a large variety of pro- and anti-inflammatory cytokines, chemokines and growth factors were assessed from the blood. Importantly, and in line with our above-described preclinical studies, an increased mental stress load, indicated by higher depression, psychosocial stress, and pain scores and decreased social functioning, also in humans was associated with higher TH protein expression (Fig. 1a, b) in the fracture hematoma and systemic inflammation markers (Supplementary Table 1). Importantly, TH expression in the hematoma was further co-localized with CD16 staining, indicating mainly neutrophils expressing local TH (Fig. 1c). In line with these findings, our follow-up study revealed significant positive correlations between local TH levels in the fracture hematoma at the time of surgery and the degree of mobility limitation documented at 6, 9 and 12 months post surgery, as well as the healing process impairment documented at 9 months post surgery using visual analog scales (VASs) (Fig. 1d). As further decreased psychosomatic health scores at the time of surgery correlated with pain scores rated 3, 6, 9 and 12 months post surgery (Fig. 1e), our clinical data overall support the hypothesis that mental trauma load also in humans is associated with a misbalanced inflammation and an increased myeloid CA production

capacity locally in the fracture hematoma, which in turn delays bone healing and increases pain nociception. Notably, other factors known to influence fracture healing, namely age, gender, body mass index, smoking, diabetes and alcohol consumption (Supplementary Table 5) did not correlate with TH expression in the fracture hematoma.

### TH KO in myeloid cells protects against negative stress effects on bone growth and regeneration

To investigate whether a specific TH KO in myeloid cells is protective against CSC-induced disturbance of bone metabolism and regeneration, we exposed 4 sets of TH[flox]/Cre[−] or TH[flox]/Cre[+] mice to 19 d of CSC or respective SHC conditions. To assess CSC effects on bone metabolism, one set of mice was euthanized immediately after termination of CSC on Day 20. To assess the effects on fracture healing, 3 sets of mice underwent femur osteotomy on Day 20 and were euthanized 1 d, 10 d or 21 d post-fracture. Genotyping of mouse ear punches confirmed that TH[flox]/Cre[-] but not TH[flox]/Cre[+] mice lack the CD11b-Cre PCR product (Supplementary Fig. 1a), and that WT TH[+/+] in contrast to TH[flox/+] and TH[flox/flox] mice show no floxed TH allele PCR product (Supplementary Fig. 1b). In confirmation of a successful TH KO in CD11b[+] cells from TH[flox]/Cre[+] mice, isolated CD11b[+] BM cells from the latter showed a by 90% reduced TH mRNA expression compared with TH[flox]/Cre[-] mice (Supplementary Fig. 1c), while the PCR product for the deleted TH allele was only detectable in BM CD11b[+] cells from TH[flox]/Cre[+] but not from TH[flox]/Cre[-] mice (Supplementary Fig. 1d). Moreover, one set of C57BL/6N WT mice was exposed to 19 d of CSC or SHC for subsequent flow cytometric analysis of TH[+] BM cell subpopulations (Fig. 2m–q). Another two sets of WT mice were subjected to only 7 d of CSC (Fig. 2a–l) or to 19 d of CSC followed by 21 d of single housing (SH) (Fig. 2r–y), to investigate both early and long-lasting effects of CSC exposure. In support of CSC to reliably induce a PTSD-like phenotype in both genotypes of the TH[flox]/Cre mouseline, CSC-exposed TH[flox]/Cre[-] and TH[flox]/Cre[+] mice showed an increased anxiety-related behavior, which is in line with previous studies[19]. The latter was indicated by a lower distance moved during open field (OF; Supplementary Fig. 2a) and novel object (NO; Supplementary Fig. 2d) exploration, as well as a decreased number of entries into (Supplementary Fig. 2b) and time spent in the inner zone of the OF (Supplementary Fig. 2c). Number of entries into (Supplementary Fig. 2e) and time spent in the contact zone (Supplementary Fig. 2f) during NO exploration were not affected. Moreover, compared with respective SHC mice, both TH[flox]/Cre[-] and TH[flox]/Cre[+] mice exposed to CSC developed transient adrenal enlargement (Supplementary Fig. 3), which is the most predictive biomarker for classification and class prediction in the CSC paradigm[20] and a typical sign of chronic stress[19]. Interestingly, only SHC and CSC mice of the TH[flox]/Cre[-] but not TH[flox]/Cre[+] group showed the typical increase in plasma NE concentrations in response to femur osteotomy (Supplementary Fig. 3b) reported in other studies[21,22], suggesting that post-fracture changes in systemic NE levels are mainly mediated by myeloid cells. In line with the latter, several types of immune cells, including neutrophils and macrophages, increase CA production in response to inflammatory stimuli like lipopolysaccharide (LPS)[23]. In contrast, we did not detect any differences in epinephrine (EPI) and dopamine (DOP) levels between the groups (Supplementary Fig. 3c, d). Own previous results obtained using immunohistochemical double staining revealed that the TH signal in the fracture hematoma of CSC mice is strongly co-localized with the Ly6G[+] signal[18]. In line with these findings, flow cytometry in the present study revealed an increased percentage of TH[+]Ly6G[+] neutrophils (Fig. 2n) in isolated BM cells from CSC vs. SHC WT mice (Fig. 2m), while the percentages of TH[+]CD8[+] cytotoxic T cells (Fig. 2o) and TH[+]CD4[+] T helper cells (Fig. 2p) were not affected by CSC. The percentage of TH[+]F4/80[+] macrophages was even lower in CSC vs. SHC WT mice (Fig. 2q). Of note in this context are own earlier data showing Ly6G[+] neutrophils, amongst other myeloid cells, to be increased in the BM of non-fractured WT CSC mice compared to

**a**

| | PHQ15 Som | GAD7 Anx | PHQ9 Depri | PHQS Stress | SF36 socfunct | SF36 pain | CTQ sum |
|---|---|---|---|---|---|---|---|
| TH score hematoma | r = 0.01573 Spearman ns | r = 0.4478 Spearman ns (0.0624) | r = 0.4908 Spearman * | r = 0.5969 Spearman ** | r = 0.4928 Spearman * | r = -0.4871 Spearman * | r = -0.0479 Spearman ns |

**b**

stress score: 0     stress score: 5
TH DAPI

**c**

stress score: 0     stress score: 5
TH CD16

**d**

| | mobility impairment 3 months | progress impairment 3 months | mobility impairment 6 months | progress impairment 6 months | mobility impairment 9 months | progress impairment 9 months | mobility impairment 12 months | progress impairment 12 months |
|---|---|---|---|---|---|---|---|---|
| TH score hematoma | r = 0.4164 Spearman ns | r = 0.2587 Spearman ns | r = 0.5404 Spearman * | r = 0.4592 Spearman ns | r = 5718 Spearman * | r = 0.5638 Spearman * | r = 0.4796 Spearman * | r = 0.08474 Spearman ns |

**e**

| | PHQ15 Som | GAD7 Anx | PHQ9 Depri | PHQS Stress | SF36 socfunct | SF36 pain | CTQ sum |
|---|---|---|---|---|---|---|---|
| SF36 pain 3 months | r = -0.7625 Spearman *** | r = -0.5341 Spearman * | r = -0.4989 Spearman ns (0.0508) | r = -0.4467 Spearman ns (0.0838) | r = -0.2732 Spearman ns | r = 0.4064 Spearman ns | r = 0.01545 Spearman ns |
| SF36 pain 6 months | r = -0.5223 Spearman * | r = -0.5337 Spearman * | r = -0.4854 Spearman ns (0.0582) | r = -0.4644 Spearman ns (0.0712) | r = -0.2931 Spearman ns | r = 0.6064 Spearman * | r = -0.1571 Spearman ns |
| SF36 pain 9 months | r = -0.4032 Spearman ns | r = -0.5069 Spearman * | r = -0.4207 Spearman ns | r = -0.4193 Spearman ns (0.0966) | r = -0.2866 Spearman ns | r = 0.6236 Spearman * | r = -0.1034 Spearman ns |
| SF36 pain 12 months | r = -0.5227 Spearman * | r = -0.4775 Spearman ns (0.0632) | r = -0.5082 Spearman * | r = -0.3856 Spearman ns | r = -0.2314 Spearman ns | r = 0.4193 Spearman ns | r = 0.08106 Spearman ns |

respective SHC mice, while Ly6G[+] neutrophils in fractured WT CSC mice are increased in the fracture hematoma but not in the BM of the intact femur[18]. This redistribution of Ly6G[+] neutrophils to the fracture site as well as the accompanying deficits in fracture healing and a misbalanced local inflammation could be prevented in CSC mice by a single injection of propranolol prior to fracture[18]. Together, these data support the hypothesis that while CSC-induced BM myelopoiesis increases the percentage of various BM myeloid cells, TH protein expression is exclusively upregulated in newly generated neutrophils. The latter leave the BM in case of a fracture and immigrate into the fracture hematoma in a β-AR-dependent manner. Of note, own earlier studies revealed comparable BM CD3[+], CD4[+] and CD8[+] cell counts between intact CSC and SHC mice. Moreover, the CSC-induced reduction in the percentage of CD8[+] T cells in the fracture hematoma 24 h after femur osteotomy simply seems to reflect an indirect effect, resulting from the pronounced CSC-induced BM neutrophil proliferation and subsequent recruitment of these newly formed myeloid cells into the fracture hematoma[18]. Together with the findings obtained in TH[flox]/Cre[+] mice of the current study, which lack TH only in cells of the innate immune system (i.e., CD11b[+] myeloid cells) and

**Fig. 1 | Association of different aspects of psychosomatic health with tyrosine hydroxylase (TH) expression in the human fracture hematoma and outcome after upper ankle fracture. a** Correlational analyses of TH immunoreactivity in the fracture hematoma of human patients suffering upper ankle fracture with different aspects of psychosomatic health assessed by established questionnaires (somatic symptoms: Som, PHQ15; anxiety: Anx, GAD7; depression: Depri, PHQ9; stress load: Stress, PHQS; social functioning: socfunct, SF36; pain: SF36; childhood adversity: CTQ sum) on the days after the surgery. **b** Representative images (scale bars: 50 μm) visualizing TH immunofluorescent stainings in the fracture hematoma of a patient with low (0; left panel) and high (5; right panel) stress scores. $n_{\text{human patients}} = 20–21$. **c** Representative images (scale bars: 50 μm) visualizing TH and CD16 immuno-fluorescent double-stainings in the fracture hematoma of a patient with low (0; left panel) and high (5; right panel) stress scores. Fluorescent channels were adjusted equally for both groups. **d** Correlational analyses of TH immunoreactivity in the fracture hematoma at the day of surgery with the degree of mobility limitation, as well as the healing process impairment documented both at 3, 6, 9 and 12 months post surgery using visual analog scales (VASs). **e** Correlational analyses of different aspects of psychosomatic health assessed by established questionnaires (somatic symptoms: Som, PHQ15; anxiety: Anx, GAD7; depression: Depri, PHQ9; stress load: Stress, PHQS; social functioning: socfunct, SF36; pain: SF36; childhood adversity: CTQ sum) on the days following surgery with pain assessed by SF36 questionnaire at 3, 6, 9 and 12 months post surgery. Spearman correlation analyses, $*P \leq 0.05$; $**P \leq 0.01$, $***P \leq 0.001$ as significant correlation. $n_{\text{human patients}} = 16$ for the follow-up time points. Source data, exact n-numbers, exact p-values and used statistical tests per panel are provided in the Source Data file.

which are widely protected from the negative effects of CSC on bone metabolism and repair (Figs. 3 and 4), we hypothesize that the role of adaptive immunity in mediating the negative bone effects of CSC is rather negligible.

Support for a β-AR-dependent migration of newly formed myeloid BM cells into the fracture site is provided by flow cytometric analysis of the fracture hematoma 1 d after fracture (Fig. 4a). In line with own previous studies in WT mice[18], the fracture hematoma of CSC vs. SHC mice in the TH$^{\text{flox}}$/Cre$^-$ group was characterized by an increased percentage of CD11b$^+$ myeloid cells (Supplementary Fig. 4a), CD11b$^+$F4/80$^+$ macrophages (Supplementary Fig. 4b), CD11b$^+$Ly6C$^+$ monocytes (Supplementary Fig. 4c) and CD11b$^+$Ly6G$^+$ neutrophils (Fig. 4b) 1 d following femur osteotomy, while all these effects were absent in the TH$^{\text{flox}}$/Cre$^+$ group. Importantly, in combination with the fact that the percentage of CD11b$^+$ myeloid cells (Supplementary Fig. 4d) and CD11b$^+$Ly6G$^+$ neutrophils (Fig. 4c) in the BM were comparable between SHC and CSC mice of the TH$^{\text{flox}}$/Cre$^-$ group, but increased in CSC vs. SHC mice of the TH$^{\text{flox}}$/Cre$^+$ group, these data argue for the increased neutrophils in the fracture hematoma of CSC TH$^{\text{flox}}$/Cre$^-$ mice to originate from the BM. The effects of neutrophil-derived CAs on CD11b$^+$F4/80$^+$ macrophages (Supplementary Fig. 4e) and CD11b$^+$Ly6C$^+$ monocytes (Supplementary Fig. 4f) in the BM are more difficult to interpret, with no effects on emigration but facilitating effects on proliferation being likely. Interestingly, CSC further increased plasma concentrations of C-X-C Motif Chemokine Ligand 1 (CXCL1; Fig. 4d), a potent neutrophil chemoattractant and activator[24] mainly produced by mast cells and macrophages[25], in the TH$^{\text{flox}}$/Cre$^-$ group but decreased this factor in the TH$^{\text{flox}}$/Cre$^+$ group, always compared to respective SHC mice. Therefore, it is likely that neutrophil-derived CAs not only facilitate BM emigration of stress-induced neutrophils, but additionally promote neutrophil immigration into the fracture hematoma possibly via CXCL1.

Despite the protective effects of propranolol reported above in fractured CSC mice this already indicate a critical role of neutrophils in the CSC-induced delay of fracture healing, evidence for a critical role of neutrophil-derived CAs in CSC-induced disturbance of bone metabolism and repair is missing. Therefore, in the present study TH$^{\text{flox}}$/Cre$^+$ mice, specifically lacking the *TH* gene in myeloid cells, were exposed to either CSC alone or to CSC followed by femur osteotomy. Strikingly, while unfractured CSC vs. SHC mice in the TH$^{\text{flox}}$/Cre$^-$ group (Fig. 3a), in line with own previous studies in WT mice[15], show a reduced femur (Fig. 3b) and tibia (Fig. 3c) length as well as an increased growth plate thickness (GPT; Fig. 3d, e), trabecular tissue mineral density (Tb.TMD; Fig. 3g), trabecular thickness (Fig. 3h) and bone volume to tissue volume ratio (BV/TV; Fig. 3f, i), all these effects were absent in the TH$^{\text{flox}}$/Cre$^+$ group. Noteworthy in this context, WT mice exposed to 7 d of CSC in the current study were characterized by increased trabecular thickness (Fig. 2b), Tb. TMD (Fig. 2d), cortical TMD (Fig. 2f) and most importantly, increased BM expression of TH (Fig. 2l), while trabecular number (Fig. 2c), trabecular separation (Fig. 2e), growth plate thickness (Fig. 2g), cortical thickness (Fig. 2h), bone volume/ tissue volume

(BV/TV, Fig. 2i) and long-bone lengths (Fig. 2j, k) did not differ. Moreover, despite mice exposed to 19 d of CSC followed by 21 d of SH relative to respective SHC mice displayed a decreased Tb. TMD (Fig. 2u) and growth plate thickness (Fig. 2x), and an unaffected trabecular thickness (Fig. 2s), number (Fig. 2t) and separation (Fig. 2v), as well as BV/TV (Fig. 2w), femur length (Fig. 2y) was again reduced. Despite further experiments, especially with respect to the late time point, are required, these findings support the hypothesis that CSC-induced changes in local BM TH expression as well as other bone-related parameters occur very early during CSC exposure, while a lower long-bone length takes longer to develop but represents a long-lasting consequences of CSC. Moreover, and this is again in agreement with an own previous study in WT mice[18], the fracture callus of CSC vs. SHC mice in the TH$^{\text{flox}}$/Cre$^-$ group is characterized by a decreased percentage of Runx2 positive hypertrophic chondrocytes (Fig. 4e, f), as well as an increased number of osteoclasts per bone perimeter (N.Oc/B.Pm; Fig. 4g, i) and osteoclast surface per bone surface (Fig. 4h) 10 d after femur osteotomy, with all these effects being absent in the TH$^{\text{flox}}$/Cre$^+$ group. Total callus area (Supplementary Fig. 5a), relative bone area (Supplementary Fig. 5b), relative cartilage area (Supplementary Fig. 5c), relative soft tissue area (Supplementary Fig. 5d), number of osteoblasts per bone perimeter (N.Ob/B.Pm; Supplementary Fig. 5e) and osteoblast surface per bone surface (Ob.S/BS; Supplementary Fig. 5e) were not affected by CSC in both the TH$^{\text{flox}}$/Cre$^-$ and TH$^{\text{flox}}$/Cre$^+$ group. Interestingly, additional Collagen 10 staining in the fracture callus revealed no differences regarding hypertrophic cartilage formation between all groups (Fig. 4j). Since we also did not detect differences in general cartilage formation, these findings together with the decreased Runx2 staining in hypertrophic chondrocytes of CSC vs. SHC in the TH$^{\text{flox}}$/Cre$^-$ but not TH$^{\text{flox}}$/Cre$^+$ group suggest that the CSC-induced release of myeloid-derived CAs mainly affects the transdifferentiation of chondrocytes into osteoblasts. Support for this hypothesis is provided by the disturbed neovascularisation found in TH$^{\text{flox}}$/Cre$^-$ but not TH$^{\text{flox}}$/Cre$^+$ CSC vs. SHC mice, identified by PECAM staining in the fracture callus (Fig. 4k), as transdifferentiating chondrocytes secrete high levels of VEGF to induce blood vessel formation[26].

At the late phase of fracture healing (21 d post-fracture), CSC in the TH$^{\text{flox}}$/Cre$^-$ but not in the TH$^{\text{flox}}$/Cre$^+$ group resulted in a decreased TMD (Fig. 4l), BV/TV (Fig. 4m) and relative bone area (Supplementary Fig. 6a), as well as increased relative soft tissue area (Supplementary Fig. 6b) and relative cartilage area (Fig. 4n, o), further indicating disturbed chondrocyte-to-osteoblast transition. Further, N.Oc/B.Pm (Supplementary Fig. 6c) and Oc.S/BS (Supplementary Fig. 6d) were significantly increased only in the TH$^{\text{flox}}$/Cre$^-$ CSC group. Of note, while BMD (Supplementary Fig. 6e), total callus area (Supplementary Fig. 6f) and total callus volume (Supplementary Fig. 6g) were not affected by CSC in either group, N.Ob/B.Pm (Supplementary Fig. 6h) and Ob.S/BS (Supplementary Fig. 6i) were affected by CSC in both the TH$^{\text{flox}}$/Cre$^-$ and TH$^{\text{flox}}$/Cre$^+$ group, indicating a mechanism independent from local CAs to underlie CSC effects on osteoblast proliferation. Together, this

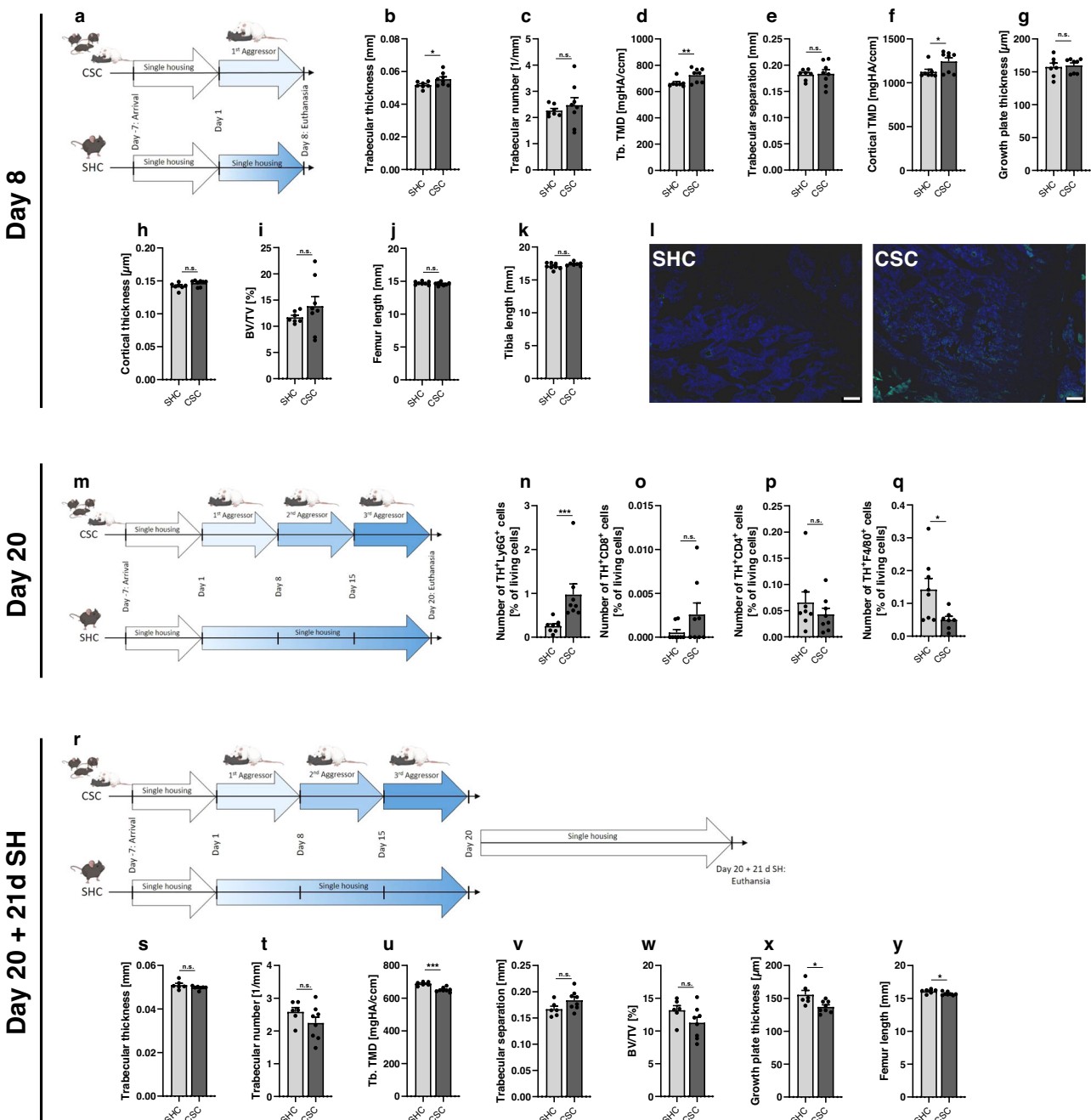

**Fig. 2 | Effects of different durations of chronic subordinate colony housing (CSC) on bone homeostasis in wild-type (WT) mice. a** Experimental timeline for WT mice exposed to 7 d of single-housed control (SHC) conditions or CSC (partly created with BioRender.com). Mice were single-housed for one week before the start of the CSC paradigm on Day 1 and euthanized on Day 8 of CSC (=7 d CSC exposure). **b** Trabecular thickness, **c** trabecular number, **d** trabecular tissue mineral density (Tb. TMD), **e** trabecular separation, **f** cortical TMD, **g** growth plate thickness, **h** cortical thickness, **i** bone volume/tissue volume (BV/TV), **j** femur length, **k** tibia length and **l** representative images of tyrosine hydroxylase (TH) immunofluorescent staining (scale bar: 100 μm) in the bone marrow (BM) of unfractured femora in WT mice euthanized on Day 8 of SHC/CSC. $n = 7$–8. **m** Experimental timeline for WT mice exposed to 19 d of SHC/CSC conditions (partly created with BioRender.com). Mice were single-housed for one week before the start of the CSC paradigm on Day 1. The aggressor mouse was changed on Days 8 and 15 and the

experimental mice were euthanized on Day 20. Number of BM **n** TH⁺Ly6G⁺ cells, **o** TH⁺CD8⁺ cells, **p** TH⁺CD4⁺ cells, and **q** TH⁺F4/80⁺ cells assessed using flow cytometry in WT mice euthanized on Day 20 of SHC/CSC exposure. $n = 7$–8. **r** Experimental timeline for WT mice exposed to 19 d of SHC/CSCconditions followed by 21 d of single housing (SH) (partly created with BioRender.com). Mice were single-housed for one week before the start of the CSC paradigm on Day 1. The aggressor mouse was changed on Days 8 and 15, and on Day 20 all experimental mice were single-housed for 21 consecutive days before being euthanized on Day 22 of SH. **s** Trabecular thickness, **t** trabecular number, **u** Tb. TMD, **v** trabecular separation, **w** BV/TV, **x** growth plate thickness, **y** femur length in WT mice exposed to 19 d of SHC/CSC + 21 d SH. $n = 6$–8. Data are presented as mean + SEM including individual values. *$P \le 0.05$, **$P \le 0.01$, ***$P \le 0.001$ versus SHC condition. n.s. not significant. Source data, exact n-numbers, exact $p$ values and used statistical tests per panel are provided in the Source Data file.

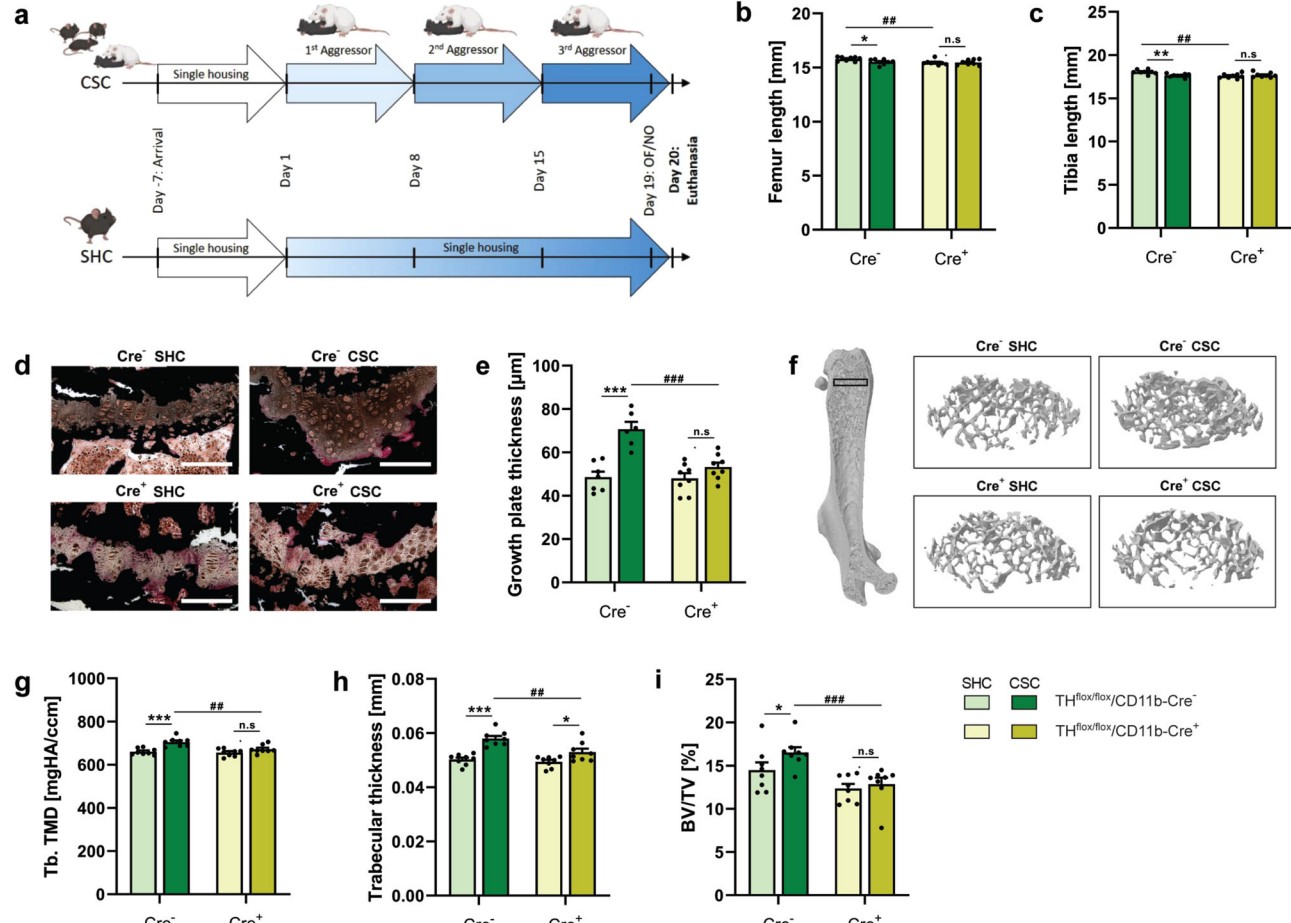

**Fig. 3 | Effects of 19 d of chronic subordinate colony housing (CSC) on bone homeostasis in TH^flox/flox^/CD11b-Cre (TH^flox^/Cre) mice. a** Experimental timeline for TH^flox^/Cre⁻ and Cre⁺ mice exposed to 19 d of single-housed control (SHC)/CSC conditions (partly created with BioRender.com licensed to SOR). Mice were single-housed for one week before the start of the CSC paradigm on Day 1. The aggressor mouse was changed on Days 8 and 15 and the experimental mice were tested for anxiety-like behavior in the open field/novel object (OF/NO) test on Day 19 and euthanized on Day 20. **b** Femur length, **c** tibia length, **d** representative images of growth plates (scale bars: 100 µm), **e** growth plate thickness, **f** representative 3D images of analyzed volume of interests (VOIs), **g** trabecular tissue mineral density (Tb. TMD), **h** trabecular thickness and **i** bone volume to tissue volume ratio (BV/TV) of unfractured femora of TH^flox^/Cre⁻ and Cre⁺ SHC/ CSC mice. *n* = 6–8. Data are presented as mean + SEM including individual values. *$P \leq 0.05$, **$P \leq 0.01$, ***$P \leq 0.001$ versus respective SHC condition; ##$P \leq 0.01$, ###$P \leq 0.001$ versus respective TH^flox^/Cre⁻ group. n.s. not significant. Source data, exact n-numbers, exact *p* values and used statistical tests per panel are provided in the Source Data file.

data strongly supports the conclusion that the negative effects of CSC on endochondral ossification both during bone metabolism and fracture healing are directly mediated by locally secreted myeloid-derived CAs. Noteworthy, myeloid cell-derived CAs also have a transient and stress-independent effect on bone homeostasis and influence the cross talk between immune and bone cells during fracture healing[27].

## CAs negatively affect in vitro chondrocyte-to-osteoblast transdifferentiation

In a next step, the effects of synthetic CAs (Fig. 5a) and conditioned medium (Fig. 5c) from isolated CD11b⁺ myeloid BM cells from SHC/CSC WT mice on chondrocyte-to-osteoblast transdifferentiation were assessed in a recently established in vitro transdifferentiation assay using the chondrogenic cell line ATDC5[28]. In agreement with the hypothesis that CSC indeed compromises chondrocyte-to-osteoblast differentiation via myeloid-derived CAs secreted locally in the BM and/ or fracture hematoma, all tested synthetic CAs (Fig. 5b) and the conditioned medium from CSC WT myeloid cells (Fig. 5d–h) inhibited the expression of characteristic pluripotency (i.e., *Sox2*, Fig. 5d; *Nanog*, Fig. 5e) and osteogenic marker (i.e., *Cbfa1*, Fig. 5f; Sp7, Fig. 5g; *Alpl*, Fig. 5h) genes in the ATDC5 cells. Of note, the effects of NE and EPI

were more prominent than the effects of DOP (Fig. 5b). Support for the negative effects of the conditioned medium from CSC WT myeloid cells on the chondrocyte-to-osteoblast transdifferentiation to be indeed mediated via CAs, all tested CA receptor antagonists were protective against the CSC-induced downregulation of pluripotency (i.e., *Sox2*, Fig. 5d; *Nanog*, Fig. 5e) and osteogenic marker (i.e., *Cbfa1*, Fig. 5f; Sp7, Fig. 5g; *Alpl*, Fig. 5h) genes in the presence of conditioned medium from CSC WT myeloid cells. To the best of our knowledge, the here employed in vitro assay is the only currently available 2D culture model in which a chondrogenic cell line can be forced to transdifferentiate into osteoblasts. While using a monolayer cell line model for sure has several drawbacks, high reproducibility and standardization, as described earlier by our group[28], represent strong and important advantages.

## β2-AR KO in chondrocytes protects against negative stress effects on bone growth

To finally confirm the critical role of chondrogenic β2-AR signaling during CSC-induced inhibtion of chondrocyte-to-osteoblast transdifferentiation under in vivo conditions, mice specifically lacking the β2-AR in chondrocytes (Adrb2^flox^/Cre⁺) were exposed to 19 d of SHC or

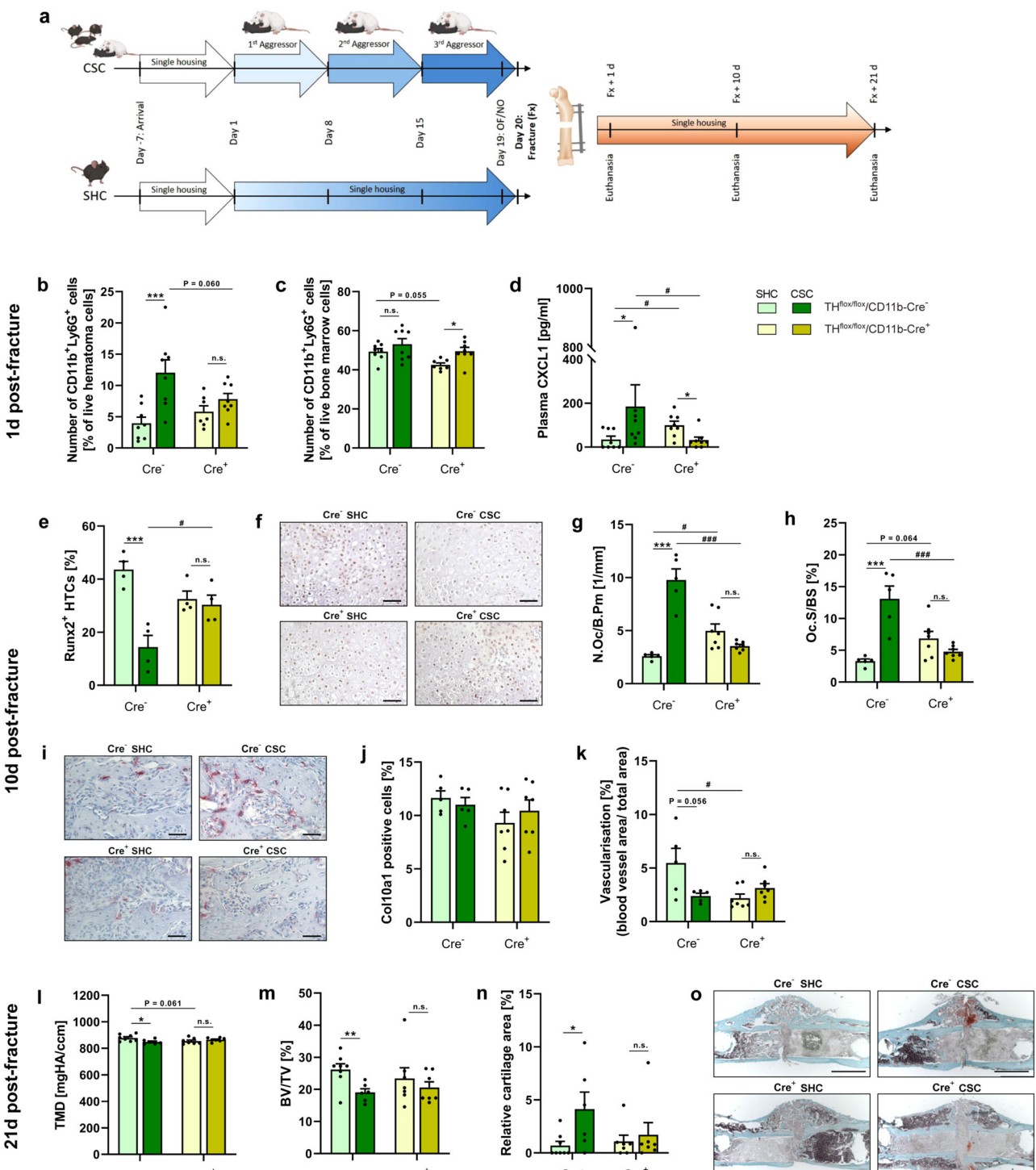

**Fig. 4 | Effects of 19 d of chronic subordinate colony housing (CSC) on fracture healing in TH^flox/flox^/CD11b-Cre (TH^flox^/Cre) mice. a** Experimental timeline for TH^flox^/Cre⁻ and Cre⁺ exposed to 19 d of single-housed control (SHC)/CSC conditions and femur osteotomy on Day 20 (partly created with BioRender.com licensed to SOR). Mice were single housed for one week before the start of the CSC paradigm on Day 1. The aggressor mouse was changed on Days 8 and 15 and the experimental mice were tested for anxiety-like behavior in the open field/novel object (OF/NO) test on Day 19. The experimental mice underwent femur osteotomy on Day 20 and were euthanized 1 d, 10 d or 21 d post surgery. Number of CD11b⁺Ly6G⁺ neutrophils in **b** the fracture hematoma and **c** the bone marrow and **d** plasma CXCL1 concentration of TH^flox^/Cre⁻ and Cre⁺ mice exposed to CSC/SHC conditions and femur osteotomy 1 d post fracture. **e** percentage of Runx2⁺ hypertrophic chondrocytes (HTCs) in the fracture callus, **f** representative immunohistochemical images of

Runx2 immunostainings in fracture callus sections (scale bars: 50 μm), **g** number of osteoclasts (N.Oc) per bone perimeter (B.Pm) and **h** osteoblast surface per bone surface (Oc.S/BS) in the fracture callus, **i** representative images of tartrate-resistant acid phosphatase (TRAP)-stained fracture callus sections (scale bars: 50 μm), **j** Col10a1⁺ cells and **k** vascularization in the fracture callus of TH^flox^/Cre⁻ and Cre⁺ SHC/CSC mice 10 d post-fracture. **l** Tissue mineral density (TMD), **m** BV/TV, **n** relative cartilage area in the fracture callus and **o** representative images of Safranin-O-stained (scale bars: 1000 μm) fracture callus sections of TH^flox^/Cre⁻ and Cre⁺ SHC/CSC mice 21 d post fracture. $n = 4$–8. Data are presented as mean + SEM including individual values. *$P \le 0.05$, **$P \le 0.01$, ***$P \le 0.001$ versus respective SHC condition; #$P \le 0.05$ versus respective TH^flox^/Cre⁻ group. n.s. not significant. Source data, exact n-numbers, exact p-values and used statistical tests per panel are provided in the Source Data file.

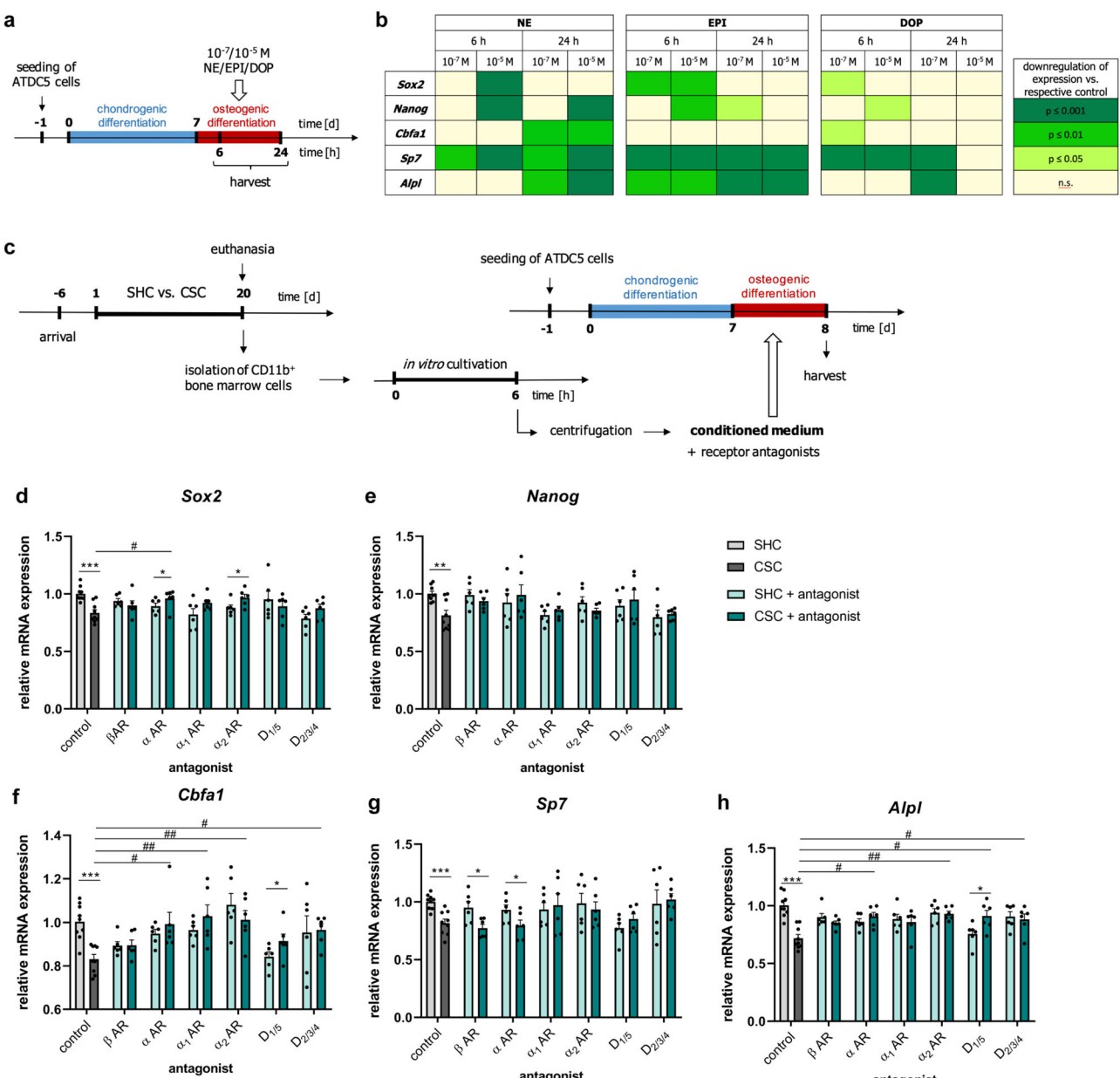

**Fig. 5 | Effect of catecholamines (CAs) or conditioned medium from single-housed control (SHC) and chronic subordinate colony housing (CSC) mice and different adrenoceptor (AR) antagonists on in vitro chondrocyte-to-osteoblast transdifferentiation. a** Experimental setup of testing different synthetic CAs on in vitro chondrocyte-to-osteoblast transdifferentiation. **b** Heatmap visualization of the effects of norepinephrine (NE), epinephrine (EPI) and dopamine (DOP) on the expression of pluripotency and osteogenic marker genes in transdifferentiating ATDC5 cells versus. respective control conditions. $n = 4$. **c** Experimental setup of testing conditioned medium of CD11b[+] myeloid bone marrow cells isolated from TH[flox/flox]/CD11b-Cre[-] and Cre[+] SHC/CSC mice in combination with antagonists for β-

Adrenoceptor (β-AR), α-Adrenoceptor (α-AR), $\alpha_1$-AR, $\alpha_2$-AR, class 1 dopaminergic receptor ($D_{1/5}$) and class 2 dopaminergic receptor ($D_{2/3/4}$) signaling. Expression of **d, e** pluripotency (**d**: Sex determining region Y box (*Sox*)*2*; **e**: *Nanog*) and **f**–**h** osteogenic marker genes (**f**: Core-binding factor alpha (*Cbfa*)*1*; **g**: Sp7; **h**: Alkaline phosphatase (*Alpl*)) in transdifferentiating ATDC5 cells after 24 h of osteogenic differentiation in conditioned medium with receptor antagonists. $n = 6$–9. Data are presented as mean + SEM. *$P \le 0.05$, **$P \le 0.01$, ***$P \le 0.001$ versus respective SHC condition; #$P \le 0.05$, ##$P \le 0.01$ versus respective control group. n.s. not significant. Source data, exact n-numbers, exact *p* values and used statistical tests per panel are provided in the Source Data file.

CSC conditions (Fig. 6a). Genotyping confirmed that Adrb2[flox]/Cre[+] but not Adrb2[flox]/Cre[-] mice show the Col2a1-Cre PCR product (Supplementary Fig. 9a), and that WT Adrb2[+/+] in contrast to Adrb2[flox/+] and Adrb2[flox/flox] mice show no floxed *Adrb2* allele PCR product (Supplementary Fig. 9b). Noteworthy in this context is that CSC-induced anxiety during OF/NO testing (Supplementary Fig. 2g-l) as well as adrenal enlargement (Fig. 6b) in Adrb2[flox]/Cre[-] mice, in contrast to WT and TH[flox]/Cre[-] mice, was not detectable, indicating genotype specific differences in stress vulnerability. This is also supported by the fact that intact bones of CSC Adrb2[flox]/Cre[-] mice (Fig. 6c–g) were only

characterized by an increased trabecular thickness (Fig. 6e) and a reduced tibia length (Fig. 6d) when compared to respective SHC mice, while all other assessed parameters were comparable between the groups. However, although CSC-induced bone effects overall were less pronounced in Adrb2[flox]/Cre[-] mice, they were absent in Adrb2[flox]/Cre[+] mice, clearly arguing for a causal role of chondrogenic β2-AR signaling in mediating the negative effects of CSC exposure on bone homeostasis. As the process of endochondral ossification during bone growth is believed to be reinitiated during fracture healing[29], likely Adrb2-KO would show an amelioratation the CSC effects on fracture

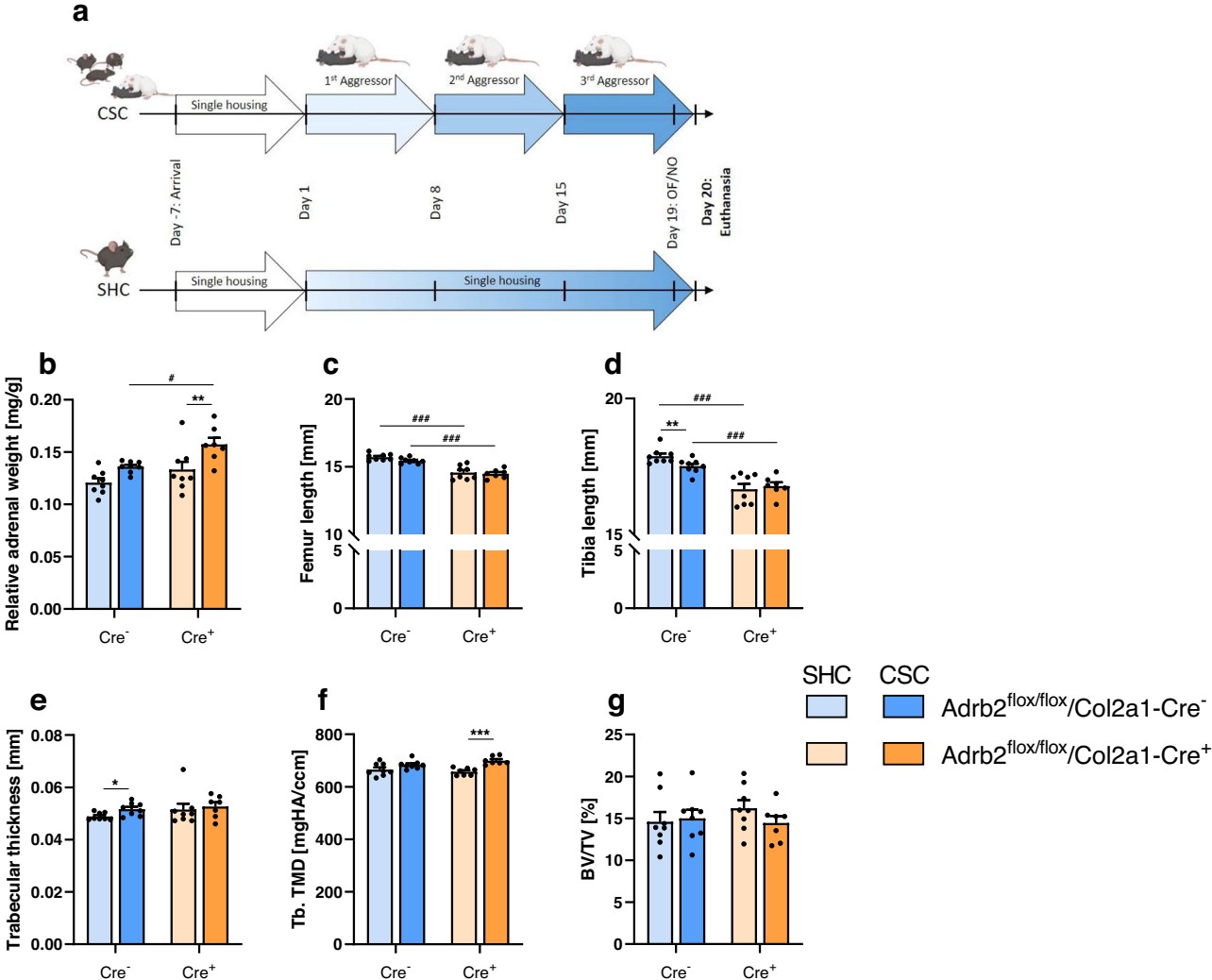

**Fig. 6 | Effects of 19 d of chronic subordinate colony housing (CSC) on bone homeostasis in Adrb2^flox/flox/Col2a1-Cre (Adrb2^flox/Cre) mice. a** Experimental timeline for Adrb2^flox/Cre⁻ and Cre⁺ exposed to 19 d of single-housed control (SHC)/ CSC conditions (partly created with BioRender.com licensed to SOR). Mice were single-housed for one week before the start of the CSC paradigm on Day 1. The aggressor mouse was changed on Days 8 and 15 and the experimental mice were tested for anxiety-like behavior in the open field/novel object (OF/NO) test on Day 19 and euthanized on Day 20. **b** Relative adrenal weight, **c** femur length, **d** tibia length, **e** trabecular thickness, **f** trabecular tissue mineral density (Tb. TMD), **g** bone volume to tissue volume ratio (BV/TV) of Adrb2^flox/Cre⁻ and Cre⁺ SHC/ CSC mice. n = 7–8. Data are presented as mean + SEM including individual values. *P ≤ 0.05, **P ≤ 0.01, ***P ≤ 0.001 versus respective SHC condition; #P ≤ 0.05, ###P ≤ 0.001 versus respective Adrb2^flox/Cre⁻ group. n.s. not significant. Source data, exact n-numbers, exact p values and used statistical tests per panel are provided in the Source Data file.

healing. Importantly, a complete stress resistance in Adrb2^flox/Cre⁺ mice can be excluded, as in this group CSC vs. SHC mice entered the inner zone of the OF less often (Supplementary Fig. 2h), which is indicative of increased anxiety-related behavior, and showed increased relative adrenal weight (Fig. 6A).

## Discussion

In the current study we first aimed to link mental health, local hematoma TH levels and the healing process in patients suffering upper ankle fracture, providing important support for the translational value of our preclinical approach. Next, we exposed C57BL/6N wild-type (WT) and TH^flox/flox/CD11b-Cre⁺ (referred to as TH^flox/Cre⁺) mice with a specific *TH* knockout (KO) in myeloid cells to either SHC/ CSC conditions alone or to SHC/CSC conditions followed by standardized femur osteotomy, to reveal that CAs secreted locally by myeloid cells are indeed critical for the negative effects of mental trauma on bone metabolism and regeneration. To further show that locally secreted CAs directly act on chondrocytes and compromise

their transdifferentiation towards osteoblasts, we employed synthetic CAs and conditioned media from myeloid cells of WT SHC/CSC mice, in the presence or absence of different CA receptor antagonists, in a novel in vitro assay, recently established by our group[28]. Finally, we exposed Adrb2^flox/flox/Col2a1-Cre⁺ (referred to as Adrb2^flox/ Cre⁺) mice and their Cre⁻ littermates (referred as Adrb2^flox/Cre⁻) with a specific *ß2-AR* KO in chondrocytes to SHC/CSC conditions to confirm the critical role of ß2-AR signaling in chondrocytes shown in our in vitro assay.

Together with own previous studies, our preclinical and in vitro data (Fig. 7) support the conclusion that while impaired mental health and stress in general promotes BM myelopoiesis, TH expression and, consequently, the capacity to produce/ secrete CAs is specifically facilitated in neutrophils. Neutrophil-derived CAs locally in the BM activate α (in vitro data)/β2 (in vitro and in vivo data)-ARs and DRs (in vitro data) on chondrocytes and, consequently, compromise their transdifferentiation into osteoblasts and, thus, bone metabolism. Neutrophil-derived CAs in an autocrine manner further promote their

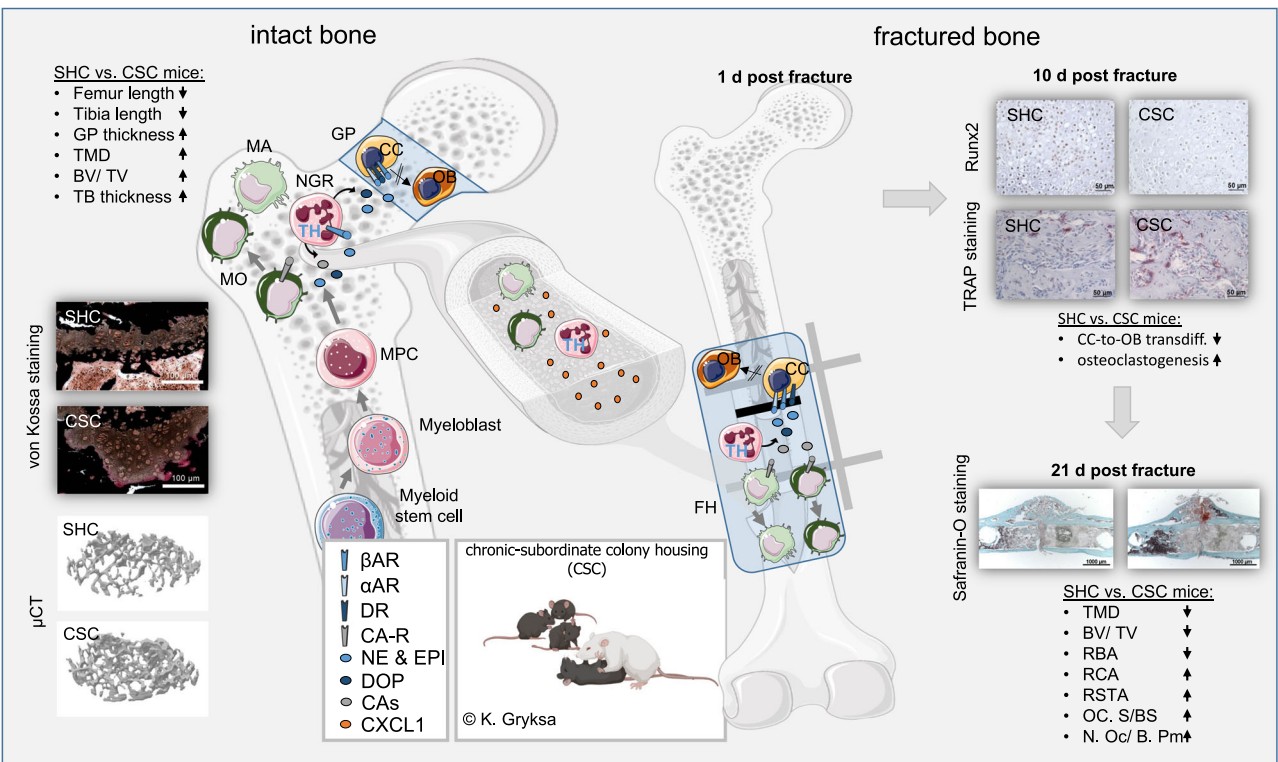

**Fig. 7 | Graphical abstract.** While mental trauma in general promotes bone marrow (BM) myelopoiesis, tyrosine hydroxylase (TH) expression and, consequently, the capacity to produce/ secrete catecholamines (CAs) is specifically facilitated in neutrophil granulocytes (NGRs). Neutrophil-derived CAs locally in the BM activate α/β-adrenoceptors (ARs) and dopaminergic receptors (DRs) on chondrocytes (CCs) and compromise their transdifferentiation into osteoblasts and, thus, bone metabolism. Neutrophil-derived CAs in an autocrine manner further promote their own BM emigration and, in case of a fracture, facilitate their own immigration into the fracture hematoma, likely in a paracrine manner by increasing CXCL1 release from hematoma mast cells and macrophages (MAs) which are two main CXCL1 producing cell types. In the fracture hematoma (FH), neutrophil-derived CAs again activate α/β-ARs and DA receptors on CCs and, consequently, compromise their transdifferentiation into osteoblasts (OBs) and, thus, adequate bone repair. (partly created with BioRender.com licensed to SOR). GP growth plate, TMD tissue mineral density, BV/TV bone volume/ tissue volume ratio, TB trabecular, GP growth plate, MPC myeloid progenitor cell, MO monocyte, RBA relative bone area, RCA relative cartilage area, RSTA relative soft tissue area, Oc.S/BS osteoclast surface/bone surface, N.Oc/ B.Pm number of osteoclasts/ bone perimeter.

own BM emigration and, in case of a fracture, facilitate their own immigration into the fracture hematoma, likely in a paracrine manner by increasing CXCL1 release from hematoma mast cells and macrophages which are two main CXCL1 producing cell types[25]. In the fracture hematoma, neutrophil-derived CAs again activate α/β2-ARs and DRs on chondrocytes and, consequently, compromise their transdifferentiation into osteoblasts and, thus, adequate bone repair. According to our clinical data, indicating an increased TH expression in fracture hematomas of patients with an increased mental stress load, which is further accompanied by a compromised fracture healing and/ or increased pain sensitivity, our preclinical data seem to be of high translational value, suggesting strategies to block immigration of TH positive myeloid cells/ neutrophils into the fracture hematoma or their local release of CAs to represent promising future strategies to facilitate fracture healing in patients who are at risk for psychosomatic disorders. Especially short-term blockade of β2-AR signaling might be useful, since several specific and unspecific β2-AR blockers with different characteristics like propranolol, solatol, atenolol, bisoprolol and metoprolol are clinically available.

## Methods

### Approval

This study complies will all relevant ethical regulations. The clinical study was approved by the Ethical Committee of the Ulm University Medical Center and conducted in accordance with the declaration of Helsinki (approval number 219/18). The mouse study was approved by the Committee on Animal Health and Care of the local government:

Regierungspräsidium Tübingen (TVAs 1195, 1216, 1219, 1267 and 1437, o135-7) and performed according to international guidelines on the ethical use of animals.

### Human samples

The clinical study was approved by the Ethical Committee of the Ulm University Medical Centre and conducted in accordance with the declaration of Helsinki (approval number 219/18). All patients gave written informed consent before study inclusion and could decide whether they agree to fill-in psychological questionnaires. Patients provided written consent to publication of their data without mentioning names. 36 patients with upper ankle fractures treated surgically at the Department of Orthopedic Trauma, Hand-, Plastic- and Reconstructive Surgery at Ulm University Medical Centre between August 2018 and August 2021 were included. Exclusion criteria were: polytrauma, pregnancy, bone diseases except primary osteoporosis, intake of bisphosphonates or parathyroid hormone, rheumatoid arthritis, open fractures of grades 3 or 4 according to Tscherne and Oestern, hepatic or renal insufficiency, cancer, intake of steroids or immunosuppressive medication, chemotherapy in the last 3 months and artificial ventilation following surgery. Fracture hematoma ($n = 21$) and venous blood in EDTA ($n = 27$) were collected during surgery, which was conducted on Days 3–5 after injury. EDTA blood samples were centrifuged to obtain plasma and stored at −80 °C until analysis. Hematoma samples were fixed in 4% formalin for 4 h and then embedded into paraffin. From the other patients enrolled in the study, we were only able to receive serum samples and therefore did not

include them into the analysis. 20 Patients agreed to additionally fill-in questionnaires one to three days after surgery. Standardized and validated questionnaires consisted of screenings from the Patient Health Questionnaire for somatic symptoms disorder (PHQ15 including symptoms for stomach pain, back pain, joint pain, menstrual pain, headaches, dizziness, heart disorders, breath shortness, problems during sexual intercourse, constipation or diarrhea, nausea, tiredness, sleep disorders, chest pain, fainting spells), depression (PHQ9), generalized anxiety (GAD-7) and psychosocial stress (PHQ-S) during the last two to four weeks, whereas higher scores indicate worse conditions[30]. Furthermore, the Short Form Health Survey (SF36)[31] was used to assess quality of life with its subscales for social functioning and pain disability (here higher scores indicate better health), and the Childhood Trauma Questionnaire (CTQ)[32] was used to self-rate maltreatment and adversity during childhood. Supplementary Table 5 summarized important patient characteristics (age, gender, BMI, smoking, alcohol consumption) from the 20 patients who were willing to fill-in the additional questionnaires at the time point of surgery. Furthermore, patient health was followed up 3, 6, 9 and 12 months after surgery. 4 patients did not agree to take part in the follow-ups at 3, 6, 9, and 12 months after the surgery. Of the 16 remaining patients, some patients missed to fill out the questionnaires at single time points. In these cases, data were replaced with last carried forward. Patient health and outcome after fracture surgery was assessed by SF36 pain questionnaire (again, higher scores indicate better health) and degree of mobility limitation as well as the healing process impairment was assessed using visual analog scales (VASs).

## Animals

Male TH$^{flox/flox}$/CD11b-Cre$^+$ mice (referred to as TH$^{flox}$/Cre$^+$ mice) of Set 1–4 with a specific *TH* knockout in myeloid cells were generated by crossing female TH$^{flox/flox}$ mice (Thtm1.1Ich), kindly provided by Prof. Dr. Ichinose, Tokyo Institute of Technology, Japan[33] with male CD11b-Cre mice (Tg(ITGAM-cre)2781Gkl), kindly provided by Dr. Vacher, Institut de Recherches Cliniques de Montréal, Québec, Canada[34]. Due to location of the CD11b-Cre construct on the Y chromosome, male TH$^{flox/flox}$/CD11b-Cre$^-$ control mice (referred to as TH$^{flox}$/Cre$^-$ mice) of Set 1–4 were generated by backcrossing female TH$^{flox/flox}$/CD11b-Cre mice with male TH$^{flox/flox}$ mice. Verification of a successful *TH* KO in CD11b$^+$ cells from TH$^{flox}$/Cre$^+$ but not TH$^{flox}$/Cre$^-$ mice was done in myeloid BM cells isolated from additional 3 mice per group (Set 5). Male C57BL/6 N (wild-type; WT) mice of Set 6 weighing 19–22 g were obtained from Charles River (Sulzfeld, Germany) to assess TH expression in various leukocyte subpopulations isolated from bone marrow. For generation of conditioned medium used in the in vitro transdifferentiation assay CD11b$^+$ BM cells were isolated from another group of WT mice exposed to 19 d of CSC or SHC (Set 7). Additional sets of mice were subjected to 7 d of CSC (Set 8) and 19 d of CSC followed by 21 d of single housing (SH; Set 9) to investigate acute and long-term effects of our stress model on the bone. Furthermore, Adrb2$^{flox/flox}$/Col2a1-Cre$^+$ mice and their Cre$^-$ littermates (referred as Adrb2$^{flox}$/Cre$^+$ and Adrb2$^{flox}$/Cre$^-$ mice, Set 10) were generated by crossing Adrb2$^{flox/flox}$ mice (Adrb2tm1Kry), generously provided by Prof. Karsenty, Department of Genetics & Development, Columbia University Medical Center, New York, USA[35], with Col2a1-Cre mice (B6;SJL-Tg(Col2a1-cre)1Bhr/J) (JAX stock #003554[36]). Genotyping of TH$^{flox}$/Cre and Adrb2$^{flox}$/Cre mice was performed using primer pairs listed in Supplementary Table 2. Male CD-1 mice (30–35 g, Charles River, Sulzfeld, Germany) were used as dominant aggressors in the CSC paradigm. All mice were kept in standard polycarbonate mouse cages (16 × 22 × 14 cm) under standard laboratory conditions (12 h light/ 12 h dark cycle, 22 °C, 60% humidity). Mice had free access to tap water and standard mouse diet (SNIFF Mausstandardfuttermittel V1534). All experiments were approved by the Committee on Animal Health and Care of the local government: Regierungspräsidium Tübingen (TVAs 1195, 1216, 1219, 1267 and 1437,

o135-7) and performed according to international guidelines on the ethical use of animals. All efforts have been made to minimize the number of animals and their suffering. Females were not used in the current study as the CSC paradigm is based on territorial aggression and the establishment of social hierarchies[19], which is not typically seen in female mice. The research described here was conducted in compliance with the ARRIVE Guidelines for Reporting Animal Research[37].

## Experimental procedures

Experimental mice of Sets 1–4, 6–10 were single-housed for one week, before they were assigned to the chronic subordinate colony housing (CSC) or single-housed control (SHC) group (Day 1, Figs. 2a, m, r, 3a, 4a, and 6a). Experimental mice of Set 8 were euthanized in the morning of Day 8 of the CSC paradigm. On Day 19 of the CSC paradigm all mice of Sets 1–4 and 10 were tested for general anxiety-like behavior in the open field/ novel object (OF/ NO) test. In the morning of Day 20 of the CSC paradigm, mice were either euthanized (Set 1, 6, 7, 10) or underwent a femur osteotomy (Sets 2–4) or were single-housed for 21 d (Set 9). Mice undergoing femur osteotomy were kept individually after the surgery and euthanized either 1 d (Set 2), 10 d (Set 3) or 21 d (Set 4) post-fracture. All experimental mice were euthanized between 06.00 and 10.00 AM by rapid decapitation following brief $CO_2$ inhalation. Of note, parameters related to CSC-induced changes in hypothalamus-pituitary-adrenal (HPA) axis activity (i.e. plasma adrenocorticotropic hormone (ACTH) and corticosterone concentrations, adrenal in vitro ACTH sensitivity, adrenal weight)[38] and the development of glucocorticoid resistance in the spleen (i.e. spleen weight, bite score, cell viability of isolated and in vitro stimulated splenocytes and relative CD11b, macrophage migration inhibitory factor (MIF) and glucocorticoid receptor (GR) protein expression in the spleen) assessed in the same experimental mice euthanized on Day 8 (Set 8) of the current study have been reported earlier by our group[38,39].

## Chronic subordinate colony housing (CSC) procedure (Sets 1–4, 6–10)

The chronic subordinate colony housing (CSC) paradigm was conducted as previously described[14,19,39–41]. Briefly, on Day 1 all experimental mice were assigned to either the CSC or the SHC group in a body weight-matched manner. Afterwards, four experimental CSC mice were housed together with a dominant CD-1 male mouse for 19 consecutive days, in order to induce a chronic stressful situation. Before the CSC procedure, the future dominant males were tested for their aggressive behavior and mice that injured their opponents by excessive aggression were excluded. Notably, the number of bite wounds received by the residents could thereby be reduced, but not totally prevented. To avoid habituation, each dominant male was replaced by a novel dominant male at Days 8 and 15 of the CSC procedure. SHC mice remained undisturbed in their home cages except for change of bedding once a week. Based on our previous data indicating pronounced social hierarchy effects on physiological and behavioral parameters when group housing non-familiar same-size male conspecifics, single housing and not group housing is considered to represent the most appropriate housing condition for controls in this paradigm[42].

## Open field/ novel object (OF/ NO) test (Sets 1–4, 10)

To assess CSC effects on anxiety-related behavior, SHC and CSC mice were exposed to the OF/ NO test on Day 19 of CSC exposure. Briefly, the arena (45 cm length × 27 cm width × 27 cm height) was subdivided into an inner (27 cm × 9 cm) and an outer zone. The arena was cleaned thoroughly before each trial. Within each trial, the mouse was placed into the inner zone and was allowed to explore the arena for 5 min. After 5 min of open-field exploration, a plastic round object (diameter: 3.5 cm; height: 1.5 cm) was placed in the middle of the inner zone. The mouse now was allowed to explore the arena containing the unfamiliar

object for 5 min. In the OF test, the number of inner zone entries and the time spent in the inner zone of the arena as well as the distance moved were assessed. In the NO test, the number and time of object explorations as well as the distance moved were analyzed. All parameters were analyzed using EthoVision XT (Version 9, Noldus Information Technology, Wageningen, Netherlands). The test was performed between 06:00 and 10:00 AM under white light conditions (350 lux).

### Femur osteotomy (Set 2–4)

On Day 20 of CSC/SHC, a standardized osteotomy of the right femur stabilized with a semi-rigid external fixator was performed as described previously[43]. Anesthesia was performed using 2% isoflurane. Mice received Tramadol (25 mg/L) in the drinking water from 1 d pre- until 3 d post surgery. All mice had a weight lost <10% of their body weight post surgery.

### Determination of adrenal glands weight (Set 1–4, 10)

After decapitation under $CO_2$ anaesthesia in the morning (between 06.00 and 10.00 a.m.) of the respective experiment, the adrenal glands were removed, pruned of fat and weighed.

### Trunk blood sampling (Sets 1–4)

Within 3 min after removing the cage from the animal room, mice were decapitated following brief $CO_2$ anaesthesia. Three droplets of trunk blood were collected in tubes containing ethylendiaminetetraacetic acid (EDTA; end concentration: 5 mM; PanReac AppliChem, ITW Reagents, Darmstadt, Germany) and sodium metabisulfite ($Na_2S_2O_5$; end concentration: 4 mM; Sigma-Aldrich, St. Louis, Missouri, USA) and stored on ice until centrifugation (for determination of plasma norepinephrine (NE), epinephrine (EPI) and dopamine (DOP) concentration). The rest of the blood was collected in EDTA-coated tubes (Sarstedt, Nuembrecht, Germany) and stored on ice until centrifugation (for determination of plasma C-X-C Motif Chemokine Ligand (CXCL) 1 concentration). All tubes were centrifuged at 4 °C (5000 × $g$, 10 min). Plasma samples for NE, EPI and DOP measurement were stored at −80 °C and plasma samples for CXCL1 measurement were stored at −20 °C until further analysis.

### Enzyme-linked immunosorbent assay (ELISA) (Sets 1–4)

Plasma norepinephrine (NE), epinephrine (EPI) and dopamine (DOP) were measured using a commercially available 3-CAT Research ELISA (LDN, Osnabrück, Germany). Plasma CXCL1 was measured using a commercially available Multiplex ELISA (Invitrogen, Waltham, MA, USA). All samples were measured as singlets.

### Flow cytometric analysis (Sets 2 and 6)

Bone marrow (BM) and fracture hematoma cells were isolated from $TH^{flox}/Cre^+$ and $Cre^-$ mice euthanized 1 d post-fracture (Set 2) and WT mice euthanized on Day 20 of the CSC paradigm (Set 6). Cells were incubated for 30 min with fluorescent labeled antibodies listed in Supplementary Table 3. Host-specific isotype controls were used as negative controls. Cells from fractured $TH^{flox}/Cre^+$ and $Cre^-$ mice were additionally stained with 7-AAD (1:200). Flow cytometric analysis was performed using LSRII flow cytometer (BD Bioscience) and BD FACS Diva software. The gating strategies used for the analysis of the flow cytometric measurements are shown in Supplementary Fig. 8.

### Magnetic activated cell sorting (MACS) separation of CD11b$^+$ bone marrow cells (Sets 5 and 7)

CD11b$^+$ bone marrow cells were isolated by MACS from $TH^{flox}/Cre^-$ and $TH^{flox}/Cre^+$ mice (Set 5) for verification of the $TH$ knockout and from SHC and CSC WT mice for the in vitro experiment (Set 7). MACS was performed using CD11b MicroBeads (Miltenyi Biotec, Bergisch Gladbach, Germany) according to the manufacturer's instructions.

Briefly, cells were incubated with CD11b MicroBeads for 10 min at 4 °C and CD11b$^+$ cells were separated from the CD11b$^-$ fraction using a Quadro-MACS separator (Miltenyi Biotec) and moisturized LS columns (Miltenyi Biotec).

### µCT analysis (Sets 1 and 4, 8–10)

Unfractured and fractured femora from mice euthanized 21 d post fracture were fixed in 4% formalin for 48 h and µCT scanning was performed using the Skyscan 1172 scanning tool (Bruker, Billerica, Massachusetts, USA) operating at 50 kV, 200 mA and a voxel resolution of 8 µm. Three-dimensional analysis was performed using CTAn and CTVol software (Bruker) according to the ASBMR guidelines[44]. The volume of interest (VOI) for unfractured femora was defined as a 280-µm-thick region in between of the cortex in a distance of 360 µm from the growth plate. The VOI for fractured femora was defined as the entire periosteal callus between the inner pinholes. Tissue mineral density was assessed using two phantoms with defined hydroxyapatite (HA) contents of 250 and 750 mg/ccm. The threshold for mineralized bone tissue was set at 394 mg HA/ccm for unfractured trabecular bone and at 642 mg HA/ccm for the fracture callus.

### Histomorphometric analysis (Set 1, 3 and 4)

Histomorphometric analysis was performed using Leica Application Suite X software (Leica, Wetzlar, Germany). Growth plate thickness of unfractured femora was measured on von Kossa-stained sections. Tissue composition of fracture calli was analyzed on Safranin-O-stained sections in a region of interest (ROI) determined as the periosteal callus between the inner pinholes together with the endosteal callus within the fracture gap. Osteoblasts and osteoclast numbers in the fracture callus were determined on Osteocalcin- and TRAP-stained sections using Osteomeasure software (OsteoMetrics, Decatur, GA, USA).

### Immunohistochemistry (mouse Sets 3, 4, 8 and human samples)

Immunohistochemical staining against Osteocalcin and Runx2 was performed using following primary antibodies incubated overnight at 4 °C: rabbit-anti-Runx2 (1:50, #8486, Cell Signaling, Danvers, Massachusetts, USA) and rabbit-anti-Osteocalcin (1:200, #orb77248, Biorbyt, Cambridge, UK). The secondary antibody biotin-XX-goat anti-rabbit (1:200, #B2770, Life technologies, Carlsbad, CA, USA) was applied for 1 h at RT, followed by 30 min incubation with horseradish peroxidase conjugated streptavidin (#PK-6100, VECTASTAIN Elite ABC-HRP Kit, Vector Laboratories, Burlingame, UK) and with NovaRED substrate (#SK-4800, Vector NovaRED Substrate Kit, Vector Laboratories) for 2–5 min. Sections were counterstained with hematoxylin (1:2000, Waldeck, Münster, Germany) and analyzed with Leica Application Suite X software. Human samples were stained for TH by using the rabbit-anti-human TH antibody (1:50, Merck #AB152) overnight and then visualizing the primary antibody by using an AF594-conjugated goat anti-rabbit antibody (1/200) for 1 h at room temperature. DNA was stained with Hoechst for 1 min. Double staining for TH and CD16 was done with the above-mentioned TH staining protocol and using an PerCP-labeled anti-CD16 antibody (1:50, Biolegend 302030). Immunohistochemical staining against PECAM (CD31, endothelial cells) and collagen X (hyertrophic chondrocytes) was performed using following primary antibodies incubated overnight at 4 °C: rat anti-CD31 (1:10, #DIA-310, Dianova) and rabbit-anti-colX (1:200, #ABIN1077945, Antibodies Online). The secondary antibody biotin-XX-goat anti-rabbit (1:200, #B2770, Life technologies, Carlsbad, CA, USA) and biotin-goat anti-rat (1:100, Invitrogen) was applied for 1 h at RT, followed by 30 min incubation with horseradish peroxidase conjugated streptavidin (#PK-6100, VECTASTAIN Elite ABC-HRP Kit, Vector Laboratories, Burlingame, UK) and with NovaRED substrate (#SK-4800, Vector NovaRED Substrate Kit, Vector Laboratories) for 3–6 min. Sections were

counterstained with hematoxylin (1:2000, Waldeck, Münster, Germany) and analyzed with Leica Application Suite X software. Negative controls with spezies-specific IgG were used to confirm specific staining (Supplementary Fig. 7).

## Cell culture
An in vitro chondrocyte-to-osteoblast transdifferentiation assay using the chondrogenic cell line ATDC5 (European Collection of Authenticated Cell Cultures (ECACC)) was performed as described previously[28]. Synthetic L-norepinephrine-(+)-bitartrate, (-)-epinephrine and dopamine hydrochloride (all Abcam, Cambridge, UK) were added to the osteogenic differentiation medium in concentrations of $10^{-7}$ and $10^{-5}$ M. Osteogenic differentiation was performed for 24 or 48 h respectively. Myeloid bone marrow cells isolated from SHC and CSC mice were cultured for 6 h in Alpha medium (Biochrom), containing 10% fetal calf serum, 1% L-glutamine, 1% penicillin/ streptomycin and 50 mg/ mL L-tyrosine at 37 °C and 5% $CO_2$. The cells were centrifuged for 5 min at $1000 \times g$, conditioned medium was harvested, supplemented with β-glycerol phosphate (10 mM, Sigma-Aldrich, Missouri, USA), ascorbate 2-phosphate (0.2 M), and human bone morphogenic protein 2 (100 ng/ml, Thermo Fisher Scientific, Waltham, Massachusetts, USA) for the induction of osteogenic differentiation and transferred to chondrogenically pre-differentiated ATDC5 cells. Besides osteogenic differentiation supplements, the catecholamine receptor antagonists propranolol (1 μM, Sigma-Aldrich, St. Louis, MO, USA), phentolamine mesylate (10 μM, Santa Cruz Biotechnology, Dallas, TX, USA), Terazosin hydrochloride (1 μM, Sigma-Aldrich), RX821002 hydrochloride (Sigma-Aldrich), R(+)-SCH-23390 hydrochloride (1 μM, Sigma-Aldrich) and S-(-)-eticlopride hydrochloride (1 μM, Sigma-Aldrich) were added to the conditioned medium, respectively. Osteogenic differentiation was performed for 24 h.

## Gene expression analysis
RNA was isolated from the harvested ATDC5 cells using RNeasy Mini Kit (Qiagen, Hilden, Germany), and each sample was treated with DNase (Qiagen). One-step semi-quantitative Real-Time-PCRs were performed using the SensiFAST SYBR Hi-ROX One-Step Kit (Bioline, Meridian Bioscience, London, UK) with primers listed in Supplementary Table 3 and the Real-Time PCR System QuantStudio 3 (Thermo Fisher Scientific). Relative gene expression was calculated by normalization to the housekeeping gene *B2M* and to expressions of the respective controls via the ΔΔCT method.

## Statistics
For statistical analysis and graphical illustrations GraphPad Prism (version 9.3.1, GraphPad Software, LCC) was used. Kolmogorov-Smirnov test with Lilliefors' correction was employed to test for normal distribution with a sample size > 4. Shapiro–Wilk Test was employed to test for normal distribution with a sample size <4. Outliers in normally distributed data sets were identified by Grubbs test and excluded from further analysis (one outlier was removed in the CSC group in the number of TH$^+$Ly6G$^+$ cells (Fig. 2n) and in the SHC group in the number of TH$^+$CD4$^+$ cells (Fig. 2p); one outlier was removed in the SHC TH$^{flox}$/Cre$^+$ group in the number of CD11b$^+$Ly6G$^+$ cells in both the hematoma (Fig. 4b) and in the number of CD11b$^+$ cells in the bone marrow (Supplementary Fig. 4d), respectively; one outlier was removed in the CSC TH$^{flox}$/Cre$^+$ group in the distance moved during OF conditions (Supplementary Fig. 2a); one outlier was removed in the SHC TH$^{flox}$/Cre$^-$ group in the distance moved during NO conditions (Supplementary Fig. 2d) and the entries into the contact zone (Supplementary Fig. 2e); one outlier was removed in the tibia length of CSC mice on Day 8 (Fig. 2k); one outlier was removed in the trabecular thickness of CSC mice on Day 20 + 21 d of single housing (SH; Fig. 2s); one outlier was removed in the SHC Adrb2$^{flox}$/Cre$^+$ group in the trabecular tissue mineral density (Tb. TMD; Fig. 6f)). Normally distributed

data sets were analyzed by parametric statistics, i.e. two-tailed Student's t-tests (one factor, two independent samples) or two-tailed Student's *t*-tests with Welch's correction when appropriate and two-way ANOVA (two factors, two independent samples). For tests considering more than two samples, a significant main effect was followed by *post-hoc* analysis using Bonferroni pairwise comparison. Not normally distributed data sets were analyzed by non-parametric statistics, i.e. Mann–Whitney *U* test (MWU, one factor, two independent samples) and Kruskal–Wallis test (KW, one factor, more than two independent samples). Correlational analyses were performed using Spearman correlation (at least one parameter not normally distributed). Tests comparing more than two samples were followed by *post-hoc* Dunn's multiple comparison, when a significant main effect was found. Data are presented as bars (mean + SEM) with individual values. The level of significance was set at $P \le 0.05$. Sample size calculation was done prior to the study by using G power software (Universität Düsseldorf, Germany). Exact *n* numbers, *p* values and statistical tests for each figure panel are included into the source data file.

## Reporting summary
Further information on research design is available in the Nature Portfolio Reporting Summary linked to this article.

## Data availability
All data generated in this study are provided in the source data file. Source data are provided with this paper.

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

## Acknowledgements

The authors thank P. Hornischer, U. Binder, Y. Aydogdu, K. Karremann, A. Bömler, T. Hieber, B. Herde and I. Baum for their technical assistance and help in performing the experiments. Furthermore, the authors would also like to thank Dr. S. Ott, Dr. S. Stein, E. Merkel, S. Brämisch, R. Richter, J. Jucha, J. Derksen and E. Müller (local animal research center) for their excellent support in terms of animal housing. The presented work in this article was supported by the Collaborative Research Centre CRC1149 funded by the Deutsche Forschungsgemeinschaft, German Research Foundation, Project number 251293561 (to M.H.L. and S.O.R.). Parts of the graphical abstract and of Figs. 2, 3, 4 and 6 as well as Supplementary Fig. 8 were created using Servier Medical Art (smart.servier.com) and/or Biorender (Biorender.com; licensed to SOR). We thank Prof. Karsenty for providing the Adrb2flox mice.

## Author contributions

S.O.R., M.H.L., F.G., M.K., G.S., K.S., H.G., K.W. and A.I. planned the study; M.E.A.T.M., E.K., L.S., S.K. and M.R.K. performed the experiments; M.K., C.P. and K.S. collected the fracture hematoma from human patients; N.K. and K.W. assessed psychosomatic health status in human patients; J.V. kindly provided the CD11b-Cre mice (Tg(ITGAM-cre) 2781Gkl); H.I. kindly provided the THflox/flox mice (Thtm1.1Ich); E.K. did the statistical analysis; S.O.R., M.H.L., M.E.A.T.M. and E.K. wrote the manuscript. All authors approved the final version of the manuscript.

## Funding

## Competing interests
The authors declare no competing interests.
