## [Peer Review File · Nature Communications]

Reviewers' comments:

Reviewer #1 (Remarks to the Author):

In this manuscript, the authors describe that chronic exposure to stress compromises bone fracture healing. Stress leads to increased accumulation of neutrophils in the wound, which release catecholamines. This builds on previous work by the authors, which had shown, in a similar mouse social defeat stress model, that bone growth and healing is impaired. The authors also reported previously that neutrophils express tyrosine hydroxylase.

In the current manuscript, clinical data correlate perceived stress in patients with bone fractures to hematoma TH expression and inflammation. The authors then pursue a CD11b driven deletion of tyrosine hydroxylase, followed by femur osteotomy. TH deletion improved healing. In vitro data indicate that catecholamines compromise osteoblast differentiation. The stress model is valid and the mice were carefully geno/phenotyped.

The main surprising aspect of this work is that neutrophils and not sympathetic nerve fibers supply catecholamines. This is particularly surprising since sympathetic nerve fibers are present in the bone marrow, more active in stress, and known to influence leukocyte and progenitor cell migration and phenotypes (see work by Frenette P., Nahrendorf M). These nerve fibers are not mentioned in the manuscript, which is an omission that should be fixed.

In the same vein, it remains unclear to which degree neutrophils contribute — it would be great to examine the relative importance compared to sympathetic nerves. Given the unusual source, direct measurements of catecholamines in WT and KO bones, perhaps by mass spec, would go a long way to really quantify to which degree leukocytes provide this neurotransmitter.

Finally, increased neutrophils may affect wound healing by many mechanisms that are unrelated to catecholamine production. To convincingly prove this point, the authors should delete the respective receptor from osteoblasts or their progenitor cells.

The histology is suggestive but would be more convincing with staining controls.

It is unclear why there are only 3 main data figures but supplemental figures.

For enumeration of leukocyte subsets, flow cytometry should be used and all gating should be shown.

Reviewer #2 (Remarks to the Author):

What are the noteworthy results?

The authors Tschaffon et al present a very nice paper on how TH positively correlates with stress, depression, and pain in human fracture hematomas following upper ankle fracture. In a CSC housing, mice lacking TH in myeloid cells revealed negative effects of mental trauma on bone turnover and repair. The effects are discussed to be mediated via a local release of catecholamines by myeloid cells.

Will the work be of significance to the field and related fields?

Discussing mental health status and its impact on regeneration is of general relevance. Key will be to show that there is a causal link between such stress/mental health aspects with clinical bone regeneration. While data is presented on the mouse model side the impact on bone healing would be nice to verify in humans.

How does it compare to the established literature? If the work is not original, please provide relevant references.

To my knowledge, the work is of original nature, and I am not aware of established literature in this specific field of interest.

Does the work support the conclusions and claims, or is additional evidence needed?

The work would substantially gain from an analyses of human fracture healing from patients with different stress, depression, and pain status. While the here presented human data is in general impressive, it lacks follow-up on fracture healing outcome. Thus, it remains unclear if the differences between the groups analyzed here (somatic symptoms: Som, PHQ15; anxiety: Anx, GAD7; depression: Depri, PHQ9; stress load: Stress, PHQS; social functioning: socfunct, SF36; pain: SF36; childhood adversity:CTQ sum) are impacted in their healing outcome. Noteworthy is that a total of 36 patients with upper ankle fractures were included. In only 21 fracture hematoma and in 27 venous blood samples were collected and only 20 patients filled in the questionnaires. At least the power behind each group would be helpful to understand. How diverse are the groups in pure patient number, gender and age? What other co-variables might have impacted the association of psychosomatic health with tyrosine hydroxylase (TH) in human fracture hematoma? Maybe in the meantime, fracture healing outcome would be possible to be evaluated across these patients.

Alternatively, the work would need to be down tuned in respect to the relevance of TH on fracture healing due to psychosomatic health.

The mouse model is impressive, and the authors have to be congratulated on the study design. However, I am a bit confused if the effects reported are now mainly associated with the adaptive immunity (and eventually the altered immune experience the mouse gain in the different housings, see Fig 1F) or mainly with the innate immunity (reflected by the differences in osteoclasts and bone resorption). As presented both effects are reported but a dominant or main line of thought that would be modulated by TH?

Finally, the main claim is the effect on endochondral ossification that different stress, depression, and pain status should have via TH. In intact bones, bone length is reduced and growth plate thickness, TMD and BV/TV are increased in SHC vs CSC. Surprisingly, in bone healing TMD, BV/TV are reduced, and endochondral ossification increased. How do these findings match to the general principle postulated that endochondral ossifications are reduced under mental stress conditions via the TH expression and, consequently, the capacity to produce/ secrete catecholamines (CAs) specifically in neutrophils?

Are there any flaws in the data analysis, interpretation and conclusions?

Beside the lack of bone healing outcome data in humans and the lacking explanation why in intact TMD and BV/TV are increased while they are reduced in regeneration there were no "flaws" that I could identify in interpretation and conclusion.

Do these prohibit publication or require revision?

A serious revision could solve the above-mentioned limitations.

Is the methodology sound? Does the work meet the expected standards in your field?

The methodology is sound. Maybe it might be helpful to consider analyzing both the cartilage formation or bone mineralization front in more depth (either by μ CT data or specific histology) to allow more in-depth analyses which specific cellular processes are really hampered.

Please describe why you consider the in vitro model as novel - that became not obvious to me.

Is there enough detail provided in the methods for the work to be reproduced?

Yes.

Reviewer #3 (Remarks to the Author):

The manuscript by Tschaffon et al. reports interesting results on a negative role of mental disturbance on bone repair, suggesting that local release of catecholamines (CAs) by neutrophils, is involved in this effect. All together the results of the study suggest that, in case of a fracture, TH expression and the capacity to produce CAs is facilitated in neutrophils providing a mechanism to promote, their migration into the fracture hematoma. Although the studies performed in humans need further confirmation, it is interesting that mental trauma in humans might be associated with unbalanced inflammatory markers and CAs increase locally.

The results are interesting and might have an important translation that is partially supported by the results obtained in humans.

In addition, confirmation of a successful TH KO in CD11b+ cells from Cre+ mice was provided, as well as CSC increase in anxiety-related behavior.

The results are well described and the methods extensively reported.

The weak part of the study is that the results are only based on a model created and utilized exclusively by the authors. Several models of stress exist and all of them have some critical issues. My main criticism is therefore that the conclusions derived from the study should have been based not only on the CSC model. Moreover, only the chronic effect of the model was studied. An acute control should have been included in the study to verify the specificity of the model and, in addition, the long term effects of the trauma produced in the CSC model should have been studied in order to perform a comparison with clinical studies that evaluate long-term disturbances. These issues should be critically evaluated and the results should be more critically discussed in the manuscript.

Specific issues:

-In the abstract is written: "...TH correlates positively with acknowledged stress, depression, and pain scores in the fracture hematoma of patients suffering upper ankle fracture...." however, it should be also reported that no correlation was found for somatic symptoms, anxiety and childhood adversity.

-In the conclusions the authors write: that" strategies to block immigration of TH positive myeloid cells/ neutrophils into the fracture hematoma or their local release of CAs represent promising future strategies to facilitate fracture healing in patients who are at risk for psychosomatic disorders....".

Examples of the promising future strategies, should be added to this sentence: administration of propranolol, NSAID, other drugs blocking TH positive neutrophils into the fracture hematoma?

-In Methods how, and which "somatic symptoms (Som, PHQ15)" were evaluated should be reported.

Point-to-point response to Reviewer' comments

Reviewer #1 (Remarks to the Author):

In this manuscript, the authors describe that chronic exposure to stress compromises bone fracture healing. Stress leads to increased accumulation of neutrophils in the wound, which release catecholamines. This builds on previous work by the authors, which had shown, in a similar mouse social defeat stress model, that bone growth and healing is impaired. The authors also reported previously that neutrophils express tyrosine hydroxylase.

Answer: Reviewer #1 is right that we earlier reported on compromised bone growth and healing as a consequence of prior chronic psychosocial stress exposure. Moreover, we reported in stressed mice an enhanced tyrosine hydroxylase (TH) protein expression right below the growth plate in intact bones as well as an increased number of TH-expressing myeloid cells in the hematoma of fractured bones. However, evidence indicating that stress-induced TH expression in and subsequent catecholamine (CAs) release from myeloid cells indeed plays a causal role in mediating the stress-induced disturbance of bone metabolism and repair was still lacking. Employing a genetically modified mouse line not able to produce TH specifically in CD11b⁺-myeloid cells in the current study now explicitly reveals that CAs produced by and released from myeloid cells locally in the intact or fractured bone are indeed critical mediators of the negative stress effects on bone metabolism and repair. Furthermore, the current manuscript for the first time reports clinical data supporting the translational value of our preclinical findings into the context of human fracture healing. Considering this Reviewer's concern, the revised version of our manuscript now not only tries to link a fracture patient's stress history with its' TH expression locally in the fracture hematoma, but also to link both parameters with fracture healing outcomes during a 12 months follow-up.

In the current manuscript, clinical data correlate perceived stress in patients with bone fractures to hematoma TH expression and inflammation. The authors then pursue a CD11b driven deletion of tyrosine hydroxylase, followed by femur osteotomy. TH deletion improved healing. In vitro data indicate that catecholamines compromise osteoblast differentiation. The stress model is valid and the mice were carefully geno/phenotyped.

Answer: We thank Reviewer #1 for his positive feedback. According to the concerns raised by Reviewer #3 we even added two more data sets to the revised version of our manuscript, one illustrating the effects of a shorter (i.e. 7 days) duration of psychosocial stress on bone metabolism and another one illustrating the persistent/transient effects of 19 d of psychosocial stress followed by a recovery phase of 21 d after stressor termination (Fig. 2).

The main surprising aspect of this work is that neutrophils and not sympathetic nerve fibers supply catecholamines. This is particularly surprising since sympathetic nerve fibers are present in the bone marrow, more active in stress, and known to influence leukocyte and progenitor cell migration and phenotypes (see work by Frenette P., Nahrendorf M). These nerve fibers are not mentioned in the manuscript, which is an omission that should be fixed.

Answer: We are grateful to Reviewer #1 for raising this concern. As in our previous publications on stress-induced disruption of bone metabolism and repair, the role of these sympathetic bone marrow nerve fibers (via β 3-adrenoceptors) on bone marrow niche cells, followed by induction of hematopoietic stem cell proliferation and increased output of neutrophils and inflammatory monocytes, was discussed in detail in the initial version of our manuscript but then, due to challenging word limitations, was removed during the progress of manuscript writing. However, according to this Reviewers concern we added the following section back to the revised version of our manuscript: *“Of note in this context, norepinephrine (NE) release by sympathetic nerve fibers during chronic variable stress signals bone marrow niche cells to decrease CXCL12 levels through the β 3-adrenoceptor, resulting in increased hematopoietic stem cell proliferation and release of neutrophils and inflammatory monocytes (Heidt et al., 2014).”*

In the same vein, it remains unclear to which degree neutrophils contribute — it would be great to examine the relative importance compared to sympathetic nerves. Given the unusual source, direct measurements of catecholamines in WT and KO bones, perhaps by mass spec, would go a long way to really quantify to which degree leukocytes provide this neurotransmitter.

Answer: We agree with the reviewer that although our study clearly identifies catecholamines (CAs) released from CD11b+ myeloid cells as mediators of negative stress effects on bone metabolism and repair, the systemic and local bone CA portion produced by either myeloid cells or sympathetic nerve fibers, both under basal and stress conditions, would be of interest in the present study. In this context, data in the initial version of the manuscript already indicated that plasma norepinephrine (NE) concentrations 1 day and 10 days following femur osteotomy were 30% lower in mice depleted of TH specifically in

CD11b+ myeloid cells compared to respective control mice (SupFig 3) and that this was independent of prior stressor exposure. This clearly indicates that post-fracture changes in systemic NE levels are mainly mediated by myeloid cells and that this systemic effect is independent of prior stressor exposure. According to the concern raised by Reviewer #1 we now added respective plasma epinephrine (EPI) and dopamine (DOP) concentrations to the revised version of the manuscript, indicating that the contribution of CD11b+ myeloid cells to systemic EPI and DOP levels prior and following femur fracture is rather negligible and independent of stress (SupFig 3 C, D). As suggested by Reviewer #1 we further quantified NE, EPI and DOP levels locally in the bone marrow of stressed and unstressed (non-fractured) TH-KO and control mice, of which only NE levels were above the detection limit of our Tri-Cat ELISA (NE, EPI, DOP; Fig. 1 of rebuttal letter). However, we decided to abstain from including these data into the revised version of the manuscript, as we only had bone material left over from 3-4 mice per group. Moreover, the absolute concentrations in our samples were on average ~factor 20 and ~factor 40 lower compared with respective control and stress mice measured in the Heidt et al., 2014 study employing a Two-Cat ELISA (NE, EPI), suggesting that local bone marrow CAs should be quantified soon after termination of the experiment, as they get degraded soon.

Fig. 1: Norepinephrine levels locally in the bone marrow

Finally, increased neutrophils may affect wound healing by many mechanisms that are unrelated to catecholamine production. To convincingly prove this point, the authors should delete the respective receptor from osteoblasts or their progenitor cells.

Answer: Although our data clearly confirm the causal role of CD11b+ myeloid cell-derived CAs in mediating the negative stress effects of bone metabolism and repair, we agree with Reviewer #1 that evidence for a direct *in vivo* effect of myeloid-derived CAs on osteoblast progenitors still remains to be confirmed. Since our *in vitro* data strongly argue for a CA-mediated disturbance of chondrocyte-to-osteoblast trans-differentiation as underlying mechanism (Fig. 5), we created a novel mouse model lacking the β 2-adrenoceptor

specifically in chondrocytes and all osteoblasts derived from chondrocytes by chondrocyte-to-osteoblast trans-differentiation by crossing Col2a1-Cre mice with Adrb2^{flox/flox} mice. These mice and respective control mice were subjected to our stress model and bone metabolism was assessed. In support of our hypothesis, we indeed found that the effects of CSC on bone length and trabecular thickness were not present in Col2a2-Cre⁺/Adrb2^{flox/flox} mice (Fig 6).

The histology is suggestive but would be more convincing with staining controls.

Answer: According to this Reviewer's suggestion negative controls have been included into the revised version of our manuscript (SupFig 7) for all immunohistochemical stainings.

It is unclear why there are only 3 main data figures but supplemental figures.

Answer: Our manuscript was originally submitted to *Nature*, which only allows 3 main data figures, and then directly transferred to *Nature Communications*. However, according to this Reviewer's suggestion and in line with the requirements of *Nature Communications* the carefully revised version of our manuscript contains 7 main figures and 9 supplemental figures.

For enumeration of leukocyte subsets, flow cytometry should be used and all gating should be shown.

Answer: According to this Reviewer's suggestion our flow cytometry gating strategies have been implemented into SupFig 8 of the revised version of our manuscript.

Reviewer #2 (Remarks to the Author):

What are the noteworthy results?

The authors Tschaffon et al present a very nice paper on how TH positively correlates with stress, depression, and pain in human fracture hematomas following upper ankle fracture. In a CSC housing, mice lacking TH in myeloid cells revealed negative effects of mental trauma on bone turnover and repair. The effects are discussed to be mediated via a local release of catecholamines by myeloid cells.

Will the work be of significance to the field and related fields?

Discussing mental health status and its impact on regeneration is of general relevance. Key will be to show that there is a causal link between such stress/mental health aspects with clinical bone regeneration. While data is presented on the mouse model side the impact on bone healing would be nice to verify in humans.

Answer: We are grateful to Reviewer #2 for his overall positive feedback and his encouraging comments. Although we don't have access to X-Ray data evaluating bony bridging of the ankle fractures, we did a follow up of our patients at 3, 6, 9 and 12 months

post-surgery and added the following section to the revised version of the manuscript: *“In line with these findings, our follow-up study revealed significant positive correlations between local TH levels in the fracture hematoma at the time of surgery and the degree of mobility limitation documented at 6, 9 and 12 months post-surgery, as well as the healing process impairment documented at 9 months post-surgery using visual analog scales (VASs) (Fig. 1D). As further decreased psychosomatic health scores at the time of surgery correlated with pain scores rated 3, 6, 9 and 12 months post-surgery (Fig. 1 E), our clinical data overall support the hypothesis that mental trauma load also in humans is associated with a misbalanced inflammation and an increased myeloid CA production capacity locally in the fracture hematoma, which in turn delays bone healing and increases pain nociception. Notably, other factors known to influence fracture healing, namely age, gender, body mass index, smoking, diabetes and alcohol consumption (SupTab 5) did not correlate with TH expression in the fracture hematoma.”*

How does it compare to the established literature? If the work is not original, please provide relevant references.

To my knowledge, the work is of original nature, and I am not aware of established literature in this specific field of interest.

Does the work support the conclusions and claims, or is additional evidence needed? The work would substantially gain from an analyses of human fracture healing from patients with different stress, depression, and pain status. While the here presented human data is in general impressive, it lacks follow-up on fracture healing outcome. Thus, it remains unclear if the differences between the groups analyzed here (somatic symptoms: Som, PHQ15; anxiety: Anx, GAD7; depression: Depri, PHQ9; stress load: Stress, PHQS; social functioning: socfunct, SF36; pain: SF36; childhood adversity:CTQ sum) are impacted in their healing outcome. Noteworthy is that a total of 36 patients with upper ankle fractures were included. In only 21 fracture hematoma and in 27 venous blood samples were collected and only 20 patients filled in the questionnaires. At least the power behind each group would be helpful to understand. How diverse are the groups in pure patient number, gender and age? What other co-variables might have impacted the association of psychosomatic health with tyrosine hydroxylase (TH) in human fracture hematoma? Maybe in the meantime, fracture healing outcome would be possible to be evaluated across these patients. Alternatively, the work would need to be down tuned in respect to the relevance of TH on fracture healing due to psychosomatic health.

Answer: As described in detail in response to the previous concern raised by Reviewer #2, we did a follow up of our patients at 3, 6, 9 and 12 months post-surgery, supporting the hypothesis that a history of life stress/prevalence of depressive symptomatology not only facilitates TH expression in the fracture hematoma but also compromises fracture healing. The latter is indicated by the facts that our follow-up study revealed significant positive correlations between local TH levels in the fracture hematoma at the time of surgery and the degree of mobility limitation documented at 6, 9 and 12 months post-surgery, as well as the healing process impairment documented at 9 months post-surgery using visual analog scales (VASs). In agreement with this Reviewer’s suggestion we provide more

sociodemographic information on our patients in the revised version of the manuscript (SupTab 5), as for instance age, gender and other co-factors known to be relevant for fracture healing (i.e., smoking, body weight, diabetes and alcohol consumption). We also did correlation analysis between TH scores and these other factors and did not detect significant correlations.

The mouse model is impressive, and the authors have to be congratulated on the study design. However, I am a bit confused if the effects reported are now mainly associated with the adaptive immunity (and eventually the altered immune experience the mouse gain in the different housings, see Fig 1F) or mainly with the innate immunity (reflected by the differences in osteoclasts and bone resorption). As presented both effects are reported but a dominant or main line of thought that would be modulated by TH?

Answer: We thank Reviewer #2 for his positive feedback on our animal model and apologize for not making the point clear enough that the innate immune system seems to be the key player in mediating the negative stress effects on bone metabolism and fracture healing. To meet this Reviewer's concern, we have added the following section to the discussion of the revised version of our manuscript and hope that the prominent role of the innate immune system is emphasized adequately: *"Of note, own earlier studies revealed comparable BM CD3+, CD4+ and CD8+ cell counts between intact CSC and SHC mice. Moreover, the CSC-induced reduction in the percentage of CD8+ T cells in the fracture hematoma 24h after femur osteotomy simply seems to reflect an indirect effect, resulting from the pronounced CSC-induced BM neutrophil proliferation and subsequent recruitment of these newly formed myeloid cells into the fracture hematoma (Haffner-Luntzer et al., PNAS 2019). Together with the findings obtained in Cre+ mice of the current study, which lack TH only in cells of the innate immune system (i.e., CD11b+ myeloid cells) and which are widely protected from the negative effects of CSC of bone metabolism and repair, we hypothesize that the role of adaptive immunity in mediating the negative bone effects of CSC is rather negligible."*

For further clarification, Fig 1F (Fig 2O in the revised version of the manuscript) only presents the percentage of CD8⁺ cells, which additionally are also positive for TH, and not the overall percentage of bone marrow CD8⁺ cells. Notably, only very few CD8⁺ cells are TH positive (less than 0.005%) and this is independent of prior stressor exposure. In addition, we want to point out that all experimental groups used in the present study were housed in the same room with identical hygienic conditions, microbial inputs and immune experiences. This is important to mention in this context, since a recent study (Bucher et al., Exp MolMed 2022) suggests that immune-experience gained by different microbial housing conditions affects fracture healing.

Finally, the main claim is the effect on endochondral ossification that different stress, depression, and pain status should have via TH. In intact bones, bone length is reduced and growth plate thickness, TMD and BV/TV are increased in SHC vs CSC. Surprisingly, in bone healing TMD, BV/TV are reduced, and endochondral ossification increased. How do these findings match to the general principle postulated that endochondral ossifications

are reduced under mental stress conditions via the TH expression and, consequently, the capacity to produce/ secrete catecholamines (CAs) specifically in neutrophils?

Answer: The reviewer is right that we found increased TMD and BV/TV in the metaphyseal part of the intact bone, while these parameters were decreased in the fracture callus. However, we propose that the mechanism underlying these phenomena are identical, namely a disturbed chondrocyte-to-osteoblast trans-differentiation during endochondral ossification. In detail, we showed earlier in intact bones that longitudinal bone growth was disturbed, as indicated by an increased growth plate width, reduced expression of transdifferentiation markers in hypertrophic chondrocytes located in the growth plate and reduced long bone length. In contrast, appositional bone growth (i.e., direct bone formation by MSCs invading the empty lacunae where chondrocytes undergo apoptosis in the growth plate) seems not to be negatively affected by mental stress. This was verified previously by unaltered TUNEL staining in chondrocytes and unaltered expression of osteocalcin in osteoprogenitor cells after stress (Foertsch et al., Dis Model Mech 2018). Therefore, we hypothesize that the long bones are shorter while the relative amount of bone in the metaphyseal part is increased as an indirect effect of disturbed endochondral ossification. In comparison to long bone growth, a higher percentage of osteoblasts are derived from chondrocytes in the fracture callus, explaining why the effects of a stress-induced disturbance of chondrocyte-to-osteoblast trans-differentiation is much more severe during fracture healing, compromising also trabecular thickness and, consequently, BV/TV. In support of the hypothesis that a stress-induced disturbance of chondrocyte-to-osteoblast trans-differentiation also plays a role in mediating the negative stress effects on fracture healing, stress exposure reduced the expression of transdifferentiation markers in hypertrophic chondrocytes in the fracture callus. Both the disturbance of endochondral ossification during long bone growth and during endochondral fracture healing after stress exposure seems to be mediated by catecholamine expression in neutrophils, as our TH^{flox}/Cre⁺ KO mice were protected from these negative effects of stress exposure. To meet this Reviewer's concern and to make our argumentation more clear we added the following section to the discussion of the revised version of the manuscript: *"We showed earlier that mental traumatization in adolescent mice negatively impacts cartilage-to-bone transition during endochondral ossification in the epiphyseal growth plate, the main site of longitudinal growth of the long bones, while appositional bone growth seems to be undisturbed. In detail, CSC mice show reduced tibia and femur lengths, mineral deposition at the growth plate and Runt-related transcription factor 2 (Runx2) expression in hypertrophic chondrocytes in the growth plate, while growth plate and trabecular thickness as well as bone mineral density (BMD) were increased in CSC compared to singly housed control (SHC) mice because of the shorter long bones."*

Are there any flaws in the data analysis, interpretation and conclusions?

Beside the lack of bone healing outcome data in humans and the lacking explanation why in intact TMD and BV/TV are increased while they are reduced in regeneration there were no "flaws" that I could identify in interpretation and conclusion.

Do these prohibit publication or require revision?

A serious revision could solve the above-mentioned limitations.

Answer: We addressed the concerns raised by Reviewer #2 in detail in both the rebuttal letter and the revised version of our manuscript and hope that our revised manuscript meets the criteria required for publication in Nature communications.

Is the methodology sound? Does the work meet the expected standards in your field?

The methodology is sound. Maybe it might be helpful to consider analyzing both the cartilage formation or bone mineralization front in more depth (either by μ CT data or specific histology) to allow more in-depth analyses which specific cellular processes are really hampered. Please describe why you consider the *in vitro* model as novel - that became not obvious to me.

Answer: We agree with Reviewer #2 that we should have described the disturbed cellular processes in the fracture callus in more detail. To meet this Reviewer's concern, we performed additional staining for PECAM to identify endothelial cells and neovascularization in the fracture callus as well as Collagen 10 staining to identify hypertrophic chondrocytes in the fracture callus. The data has been included into the manuscript and the following section has been added to the revised version of the manuscript: *"Interestingly, additional Collagen 10 staining in the fracture callus revealed no differences regarding hypertrophic cartilage formation between the groups (Fig. 4J). Since we also did not detect differences in general cartilage formation, these findings together with the decreased Runx2 staining in hypertrophic chondrocytes of CSC vs. SHC in the Cre⁻ but not Cre⁺ group suggest that the CSC-induced release of myeloid-derived CAs mainly affects the transdifferentiation of chondrocytes into osteoblasts. Support for this hypothesis is provided by the disturbed neovascularization found in Cre⁻ but not Cre⁺ CSC vs. SHC mice, identified by PECAM staining in the fracture callus (Fig. 4K), as transdifferentiating chondrocytes secrete high levels of VEGF to induce blood vessel formation."*

To better describe the novelty and advantages of our *in vitro* assay we added the following lines to the revised version of our manuscript: "To the best of our knowledge, the here employed *in vitro* assay is the only currently available 2D culture model in which a chondrogenic cell line can be forced to transdifferentiate into osteoblasts. While using a monolayer cell line model for sure has several drawbacks, high reproducibility and standardization, as described earlier by our group (Tschaflon et al., 2022 Endocrine), represent strong and important advantages."

Is there enough detail provided in the methods for the work to be reproduced?

Yes.

Reviewer #3 (Remarks to the Author):

The manuscript by Tschaffon et al. reports interesting results on a negative role of mental disturbance on bone repair, suggesting that local release of catecholamines (CAs) by neutrophils, is involved in this effect. All together the results of the study suggest that, in case of a fracture, TH expression and the capacity to produce CAs is facilitated in neutrophils providing a mechanism to promote, their migration into the fracture hematoma. Although the studies performed in humans need further confirmation, it is interesting that mental trauma in humans might be associated with unbalanced inflammatory markers and CAs increase locally. The results are interesting and might have an important translation that is partially supported by the results obtained in humans.

In addition, confirmation of a successful TH KO in CD11b+ cells from Cre+ mice was provided, as well as CSC increase in anxiety-related behavior. The results are well described and the methods extensively reported.

Answer: We thank the reviewer for these positive comments on our work.

The weak part of the study is that the results are only based on a model created and utilized exclusively by the authors. Several models of stress exist and all of them have some critical issues. My main criticism is therefore that the conclusions derived from the study should have been based not only on the CSC model. Moreover, only the chronic effect of the model was studied. An acute control should have been included in the study to verify the specificity of the model and, in addition, the long-term effects of the trauma produced in the CSC model should have been studied in order to perform a comparison with clinical studies that evaluate long-term disturbances. These issues should be critically evaluated and the results should be more critically discussed in the manuscript.

Answer: The reviewer is right that our data are solely derived from the CSC model, which represents a well-characterized and internationally accepted mouse model for posttraumatic stress disorder (PTSD), without creating any signs of co-morbid depressive-like symptomatology (Slattery et al., 2012). We are convinced that the latter represents an advantage rather than a drawback for the current study, especially when considering that the main aim of the present study was to delineate the mechanisms underlying the negative consequences of chronic/traumatic psychosocial stress on bone metabolism and repair. Of note, individuals traumatized early in life are characterized by shorter stature and growth retardation, which is in line with shorter long bone growth described in CSC mice. Moreover, both CSC mice and PTSD patients are characterized by a decreased hypothalamus-pituitary-adrenal (HPA) axis, i.e. hypocorticism (Yehuda and Seckl, 2011; Reber et al., 2016), but an increased activity of the sympathetic nervous system (SNS) (Langgartner et al., 2017; Reber et al., 2007), while depressed patients rather show increased plasma cortisol concentrations (i.e. hypercorticism) (Holsboer, 2000). Patients diagnosed with depression or mice exposed to stress models resulting in a depressive-like phenotype, like the “chronic mild stress (CMS) paradigm”, further develop a bone phenotype characterized by reduced bone mass/osteoporosis (Liu and Liu, 2017; Azuma et al., 2015; Furuzawa et al., 2014; Yirmiya et al., 2006; Baldock et al., 2014).

To meet this Reviewer's concern and to make this more clear, we have added the following section to the revised version of our manuscript information: "*Preclinical studies revealed*

reduced bone mass in murine models of chronic mild stress, in which mice were subjected to a series of mild and unpredictable physical and/or psychological stressors, resulting in a depressive-like phenotype (Liu and Liu, 2017; Azuma et al., 2015; Furuzawa et al., 2014; Yirmiya et al., 2006; Baldock et al., 2014). In contrast to mouse models for depression, employing the chronic subordinate colony housing (CSC) paradigm as an acknowledged model for social stress-associated PTSD in male mice, we showed earlier that mental traumatization in adolescent mice negatively impacts cartilage-to-bone transition during endochondral ossification in the epiphyseal growth plate, the main site of longitudinal growth of the long bones, while appositional bone growth seems to be undisturbed. In detail, CSC mice show reduced tibia and femur lengths, mineral deposition at the growth plate and Runt-related transcription factor 2 (Runx2) expression in hypertrophic chondrocytes in the growth plate, while growth plate and trabecular thickness as well as bone mineral density (BMD) were increased in CSC compared to single-housed control (SHC) mice because of the shorter long bones.”

Moreover, as suggested by the reviewer we have included bone data from the more acute CSC phase and from the CSC recovery phase (Fig. 2). In detail, one additional set of WT mice was subjected to 8 days of CSC while another set of WT mice was subjected to 20 days of CSC, followed by 21 days of SHC for recovery, before euthanizing for assessment of bone metabolism (Fig. 2). Accordingly, we have added the following section to the revised version of the manuscript: *“Noteworthy in this context, WT mice exposed to 7 d of CSC in the current study were characterized by increased trabecular thickness (Fig 2B), Tb. TMD (Fig 2D), cortical TMD (Fig 2F) and most importantly, increased BM expression of TH (Fig 2L), while trabecular number (Fig 2C), trabecular separation (Fig 2E), growth plate thickness (Fig 2G), cortical thickness (Fig 2H), bone volume/ tissue volume (BV/TV, Fig 2I) and long bone lengths (Fig. 2J, K) did not differ. Moreover, despite mice exposed to 19 d of CSC followed by 21 d of SH relative to respective SHC mice displayed a decreased Tb. TMD (Fig 2U) and growth plate thickness (Fig 2X), and an unaffected trabecular thickness (Fig 2S), number (Fig 2T) and separation (Fig 2V), as well as BV/TV (Fig 2W), femur length (Fig 2Y) was again reduced. Despite further experiments, especially with respect to the late timepoint, are required, these findings support the hypothesis that CSC-induced changes in local BM TH expression as well as other bone-related parameters occur very early during CSC exposure, while a lower long-bone length takes longer to develop but represents a long-lasting consequences of CSC.”*

Specific issues:

-In the abstract is written: *“...TH correlates positively with acknowledged stress, depression, and pain scores in the fracture hematoma of patients suffering upper ankle fracture...”* however, it should be also reported that no correlation was found for somatic symptoms, anxiety and childhood adversity.

Answer: This has been changed accordingly.

-In the conclusions the authors write: *that” strategies to block immigration of TH positive myeloid cells/ neutrophils into the fracture hematoma or their local release of CAs represent promising future strategies to facilitate fracture healing in patients who are at risk for*

psychosomatic disorders....”. Examples of the promising future strategies, should be added to this sentence: administration of propranol, NSAID, other drugs blocking TH positive neutrophils into the fracture hematoma?

Answer: We think that especially short-term blockade of beta-adrenoreceptor signaling might be useful to prevent stress-related fracture healing complications, since these medications are already clinically available. To meet this Reviewer’s concern we added the following statement to the revised version of our manuscript: “Especially short-term blockade of beta-adrenoreceptor signaling might be useful, since several specific and unspecific beta-adrenoreceptor blockers with different characteristics like propranolol, solatol, atenolol, bisoprolol and metaprolol are clinically available.”

-In Methods how, and which “somatic symptoms (Som, PHQ15)” were evaluated should be reported.

Answer: We included this information into the methods section: “*PHQ-15 including symptoms for stomach pain, back pain, joint pain, menstrual pain, headaches, dizziness, heart disorders, breath shortness, problems during sexual intercourse, constipation or diarrhea, nausea, tiredness, sleep disorders, chest pain, fainting spells*”

Reviewer #1 (Remarks to the Author):

The revisions are satisfactory.

Reviewer #2 (Remarks to the Author):

thank you - all concerns have been addressed in the reviewer reply and modified manuscript. Nothing to add...

Reviewer #3 (Remarks to the Author):

I am satisfied of how the authors have addressed all my criticisms and concerns